# Ultra-High Contrast MRI: The Whiteout Sign Shown with Divided Subtracted Inversion Recovery (dSIR) Sequences in Post-Insult Leukoencephalopathy Syndromes (PILS)

**Paul Condron [1,2], Daniel M. Cornfeld [1,2], Miriam Scadeng [1,2], Tracy R. Melzer [3,4], Gil Newburn [1], Mark Bydder [1], Eryn E. Kwon [1,5], Joshua P. McGeown [1], Geoffrey G. Handsfield [1,5], Taylor Emsden [1], Maryam Tayebi [1,2], Samantha J. Holdsworth [1,2] and Graeme M. Bydder [1,6,*]**

1    Mātai Medical Research Institute, Tairāwhiti Gisborne 4010, New Zealand;
     m.scadeng@auckland.ac.nz (M.S.); gil@neuropsychiatrygilnewburn.co.nz (G.N.);
     j.mcgeown@matai.org.nz (J.P.M.); g.handsfield@auckland.ac.nz (G.G.H.);
     s.holdsworth@matai.org.nz (S.J.H.)
2    Department of Anatomy and Medical Imaging, Faculty of Medical and Health Sciences & Centre for Brain
     Research, University of Auckland, Auckland 1010, New Zealand
3    Department of Medicine, University of Otago, Christchurch 8011, New Zealand
4    New Zealand Brain Research Institute, Christchurch 8011, New Zealand
5    Auckland Bioengineering Institute, University of Auckland, Auckland 1010, New Zealand
6    Department of Radiology, University of California San Diego, San Diego, CA 92093, USA
*    Correspondence: gbydder@health.ucsd.edu

**Abstract:** Ultra-high contrast (UHC) MRI describes forms of MRI in which little or no contrast is seen on conventional MRI images but very high contrast is seen with UHC techniques. One of these techniques uses the divided subtracted inversion recovery (dSIR) sequence, which, in modelling studies, can produce ten times the contrast of conventional inversion recovery (IR) sequences. When used in cases of mild traumatic brain injury (mTBI), the dSIR sequence frequently shows extensive abnormalities in white matter that appears normal when imaged with conventional $T_2$-fluid-attenuated IR ($T_2$-FLAIR) sequences. The changes are bilateral and symmetrical in white matter of the cerebral and cerebellar hemispheres. They partially spare the anterior and posterior central corpus callosum and peripheral white matter of the cerebral hemispheres and are described as the whiteout sign. In addition to mTBI, the whiteout sign has also been seen in methamphetamine use disorder and Grinker's myelinopathy (delayed post-hypoxic leukoencephalopathy) in the absence of abnormalities on $T_2$-FLAIR images, and is a central component of post-insult leukoencephalopathy syndromes. This paper describes the concept of ultra-high contrast MRI, the whiteout sign, the theory underlying the use of dSIR sequences and post-insult leukoencephalopathy syndromes.

**Keywords:** ultra-high contrast; magnetic resonance imaging; whiteout sign; divided subtracted inversion recovery; post-insult leukoencephalopathy syndromes; $T_1$-bipolar filter; $T_1$-BLAIR; white matter disease of the brain; mild traumatic brain injury

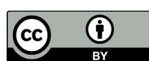

## 1. Introduction

   The first human radiograph ever taken was an image of Anna Bertha Roentgen's left hand [1] (Figure 1). It showed remarkable contrast between bone, soft tissue and the metal rings on her ring finger. Contrast on medical images is the difference in brightness (or signal) between two tissues, fluids or materials. The contrast first demonstrated by Wilhelm Roentgen on his wife's hand in 1895 had never been seen before and it became central to the practice of radiology.

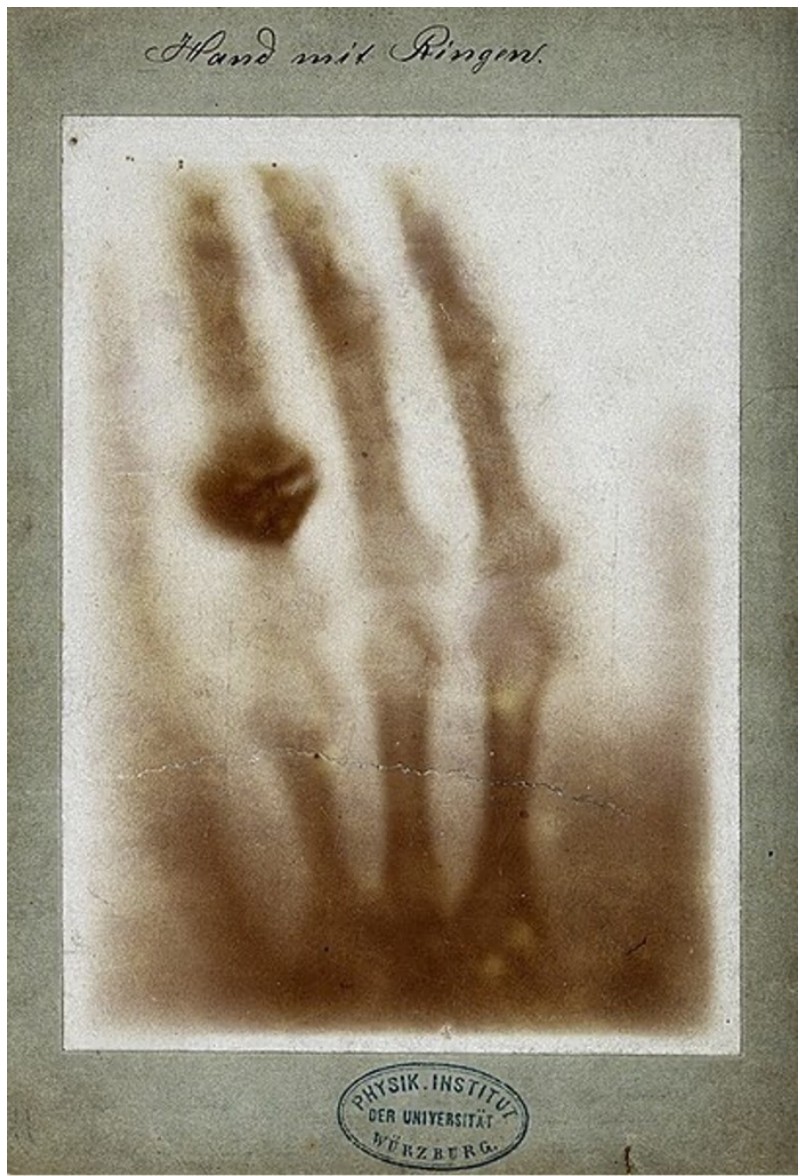

**Figure 1.** Positive radiograph of Anna Bertha Roentgen's left hand taken by Wilhelm Roentgen in December 1895 [1]. The difference in brightness (i.e., contrast) is seen between Anna's rings, the bones of her hand and soft tissues. Contrast of this type is the basis of radiology.

In plain radiographs of the head, contrast could be seen between bone of different thickness, but the soft tissue of the brain itself could not specifically be seen (Figure 2). If air was substituted for CSF, contrast could be created between the ventricular system (now containing air rather than CSF) and the surrounding brain. This technique, which was developed by Walter Dandy in 1918–1919, was used diagnostically to recognize disease of the brain by displacement or distortion of the ventricular system and/or subarachnoid space in air studies of the brain [2,3] as shown in Figure 3.

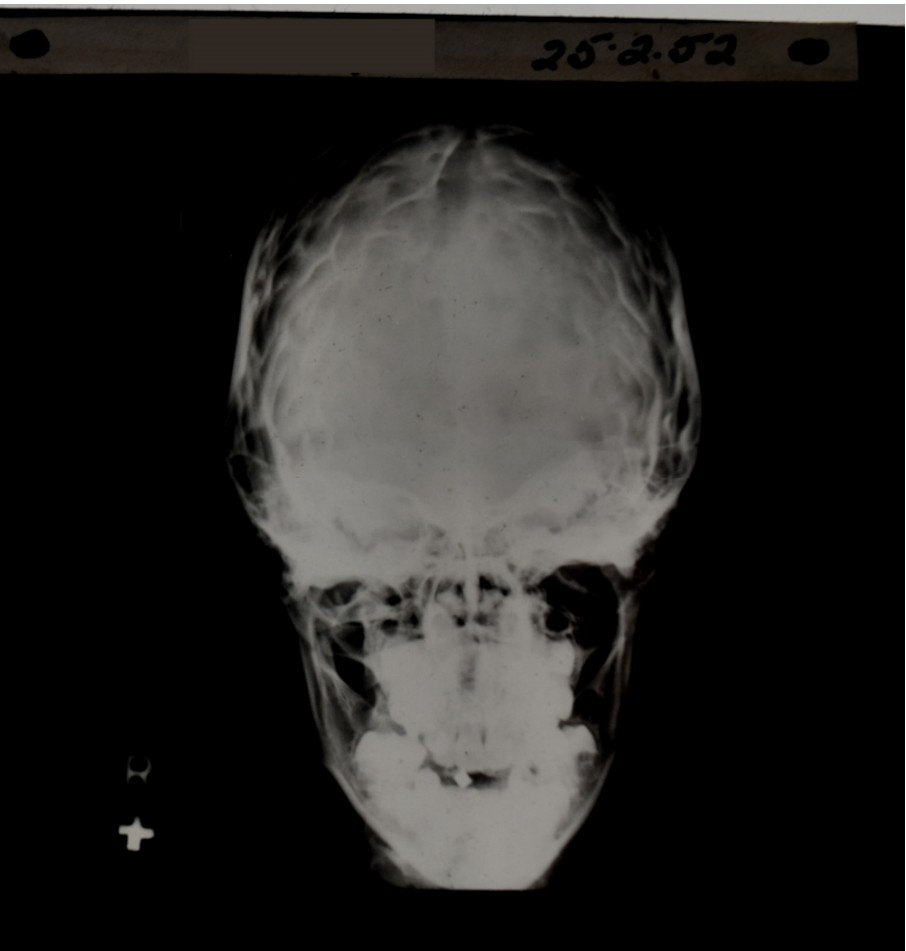

**Figure 2.** Early skull X-ray. Contrast is seen between bone and air as well as between bone of different thicknesses; however, the soft tissue of the brain is not directly observable.

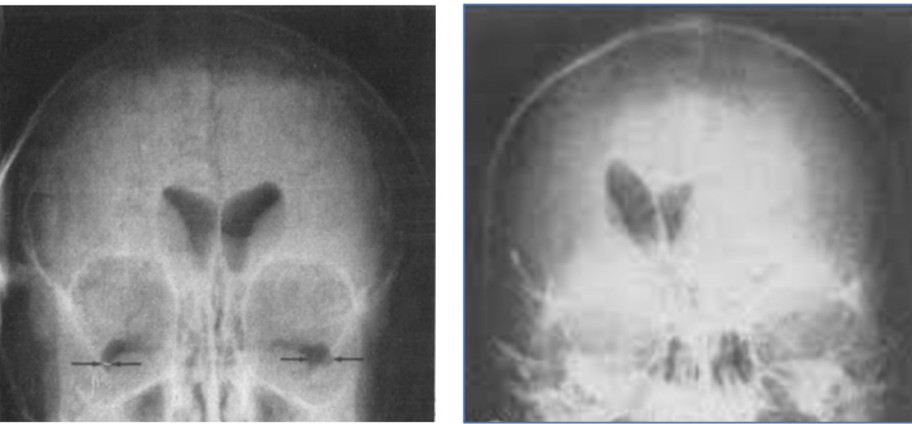

**Figure 3.** Air studies, frontal views. Normal control (**left**) and patient with brain tumor in the left cerebral hemisphere (**right**). The CSF in the ventricular has been replaced by air. The air has a low signal (black appearance) and this creates contrast between the ventricular system and surrounding brain. In the normal air study on the left, the lateral ventricles are symmetrical. In the patient on the right both lateral ventricles are displaced to the right by a tumor in the left hemisphere (black arrow). There is encroachment of the brain into the left lateral ventricle and probable enlargement of the right lateral ventricle.

Another method of creating contrast in radiographs of the brain was to inject sodium iodinate into the cerebral arteries and so create contrast between large vessels and the surrounding tissues as demonstrated by Egas Moniz in 1927 [4] and shown in Figure 4. The technique was called angiography. Obstruction of vessels, displacement of these and development of new vessels could be seen and were used in the diagnosis of disease.

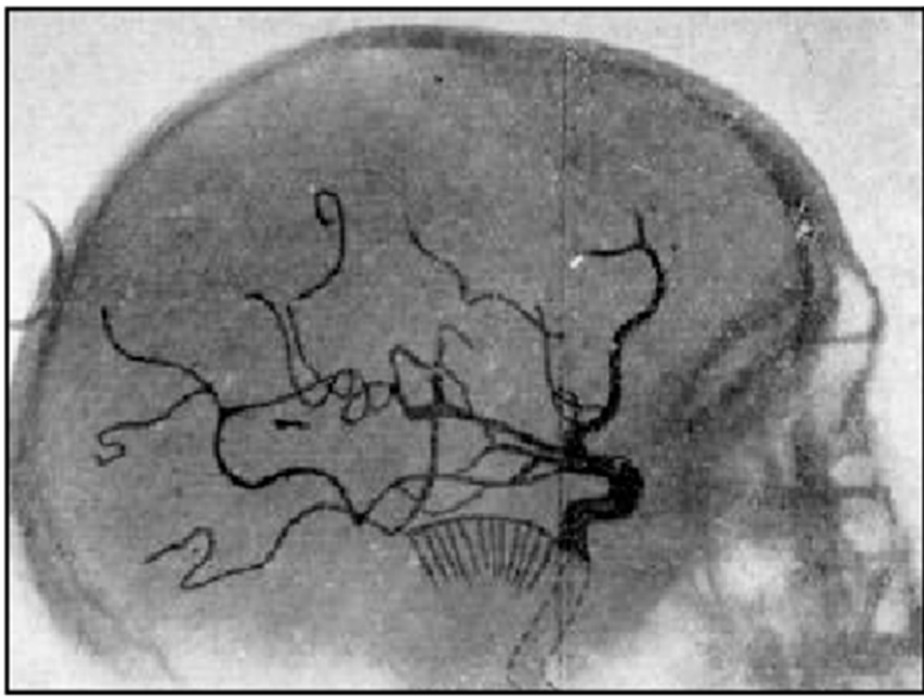

**Figure 4.** Positive radiograph of the head after arterial injection of sodium iothalamate, lateral view (1927). The sodium iothalamate is seen in larger arteries. It has a low signal, creating contrast between the arteries and the surrounding brain.

Both air studies and angiography required the introduction of a contrast agent to see disease and were only indirect methods of seeing the brain. The brain itself was not visualized with either technique. This was a problem with diseases such as multiple sclerosis (MS), where abnormal tissue could be present in the brain but not displace the ventricular system or produce vascular abnormalities. As a result, radiological diagnosis of the disease was not possible.

The first major revolution in soft tissue contrast in medical imaging came in 1971 with the introduction of brain computed tomography (CT) [5,6]. The technique not only produced slices of the brain that were much more useful than conventional tomography, but it directly displayed brain tissue. Within the brain, there was high intrinsic contrast between normal and abnormal tissues so that lesions such as the glioma shown in Figure 5 could readily be seen because its signal was lower than that of normal brain. The intrinsic contrast was present without requiring the use of a contrast agent (such as air and iodinated compounds). When intravenous iodinated contrast agents were used with CT, additional information was obtained. CT transformed the practice of neuroradiology from 1971 onwards, and body imaging from 1975 onwards as well (Figure 6).

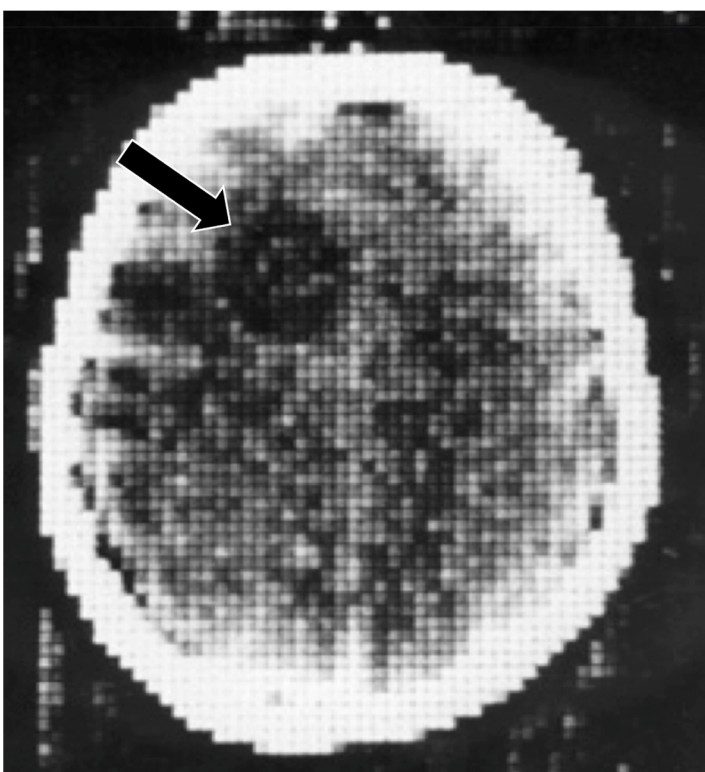

**Figure 5.** First patient examined with CT, frontal cystic glioma (October 1971) [5]. CT produced images of slices of the brain, which was a major improvement over conventional tomography. Signal is seen from the brain in this 80 × 80 matrix image. In addition, there is negative intrinsic contrast between the cystic glioma (black arrow) which is lower signal (darker) than the surrounding normal brain.

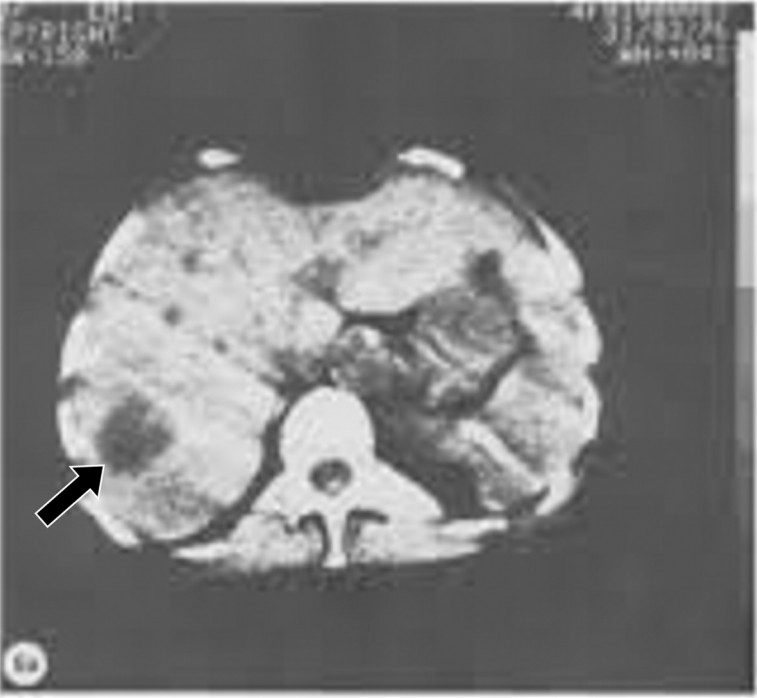

**Figure 6.** Body CT transverse image of the liver (1976). The normal liver has a moderately high signal appearance. A tumor within the liver (black arrow) has a darker appearance. There is obvious negative intrinsic contrast between the tumor and the normal liver.

The second major revolution in soft tissue contrast imaging came in 1981. While CT had shown high soft tissue contrast in the brain and elsewhere relative to plain radiographs, it was possible to obtain even higher soft tissue contrast with magnetic resonance imaging (MRI). This was seen in ten cases of MS in which 19 lesions were depicted with CT but 112 more lesions were seen with MRI using $T_1$ dependent inversion recovery (IR) sequences [7] (Figure 7). The additional lesions seen with MRI were smaller than those seen with CT and were demonstrated in white matter that appeared normal on CT. Even though MRI was much slower and of lower spatial resolution than CT, its superior soft tissue contrast resulted in a decisive advantage from a clinical point of view.

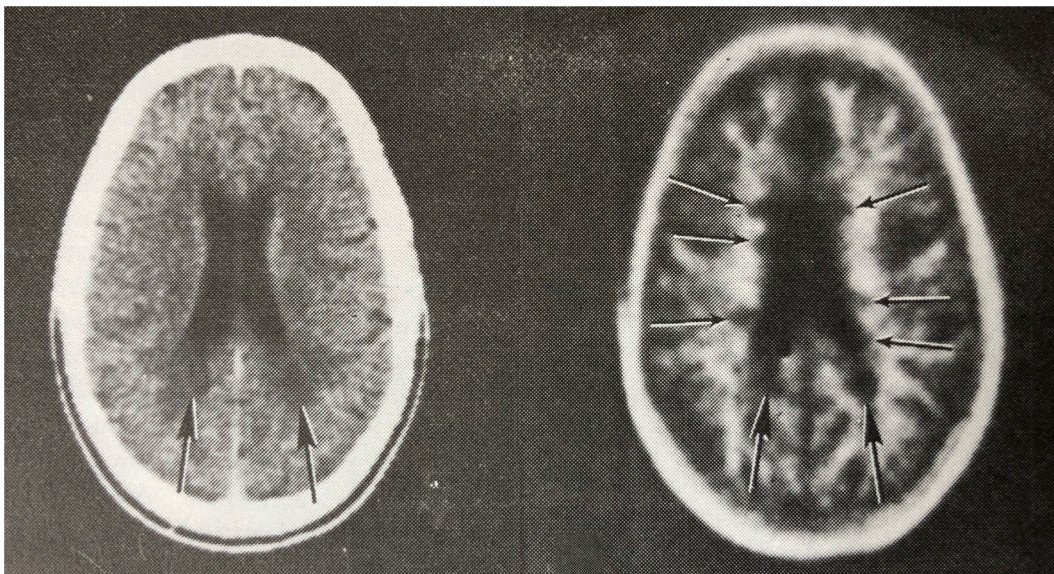

**Figure 7.** Multiple sclerosis (MS): CT (**left**) and inversion recovery (IR) (**right**) (November 1981) [7]. Two larger lesions are seen on the CT image (left, arrows). These are also seen on the IR image (right, arrows). There are an additional six lesions seen in the periventricular white matter on the IR image (arrows). The additional lesions are seen in normal appearing white matter on the CT image.

The contrast advantage of MRI was extended to included $T_2$ dependent spin echo sequences in 1982 [8,9] (Figure 8) and further improvements in the demonstration of soft tissue contrast with MRI came with short inversion time IR (STIR) (Figure 9), diffusion-weighted, susceptibility-weighting (Figure 10) and $T_2$-fluid attenuated inversion recovery ($T_2$-FLAIR) (Figure 11) sequences [10–13]. In addition, from 1984 onwards Gadolinium based contrast agents (GBCAs) were used to create additional contrast in particular clinical situations [14,15] as shown in Figure 12. Imaging protocols using multiple sequences of these and other types have been established for different clinical applications and form the basis for modern MRI examinations of the brain and other organs of the body.

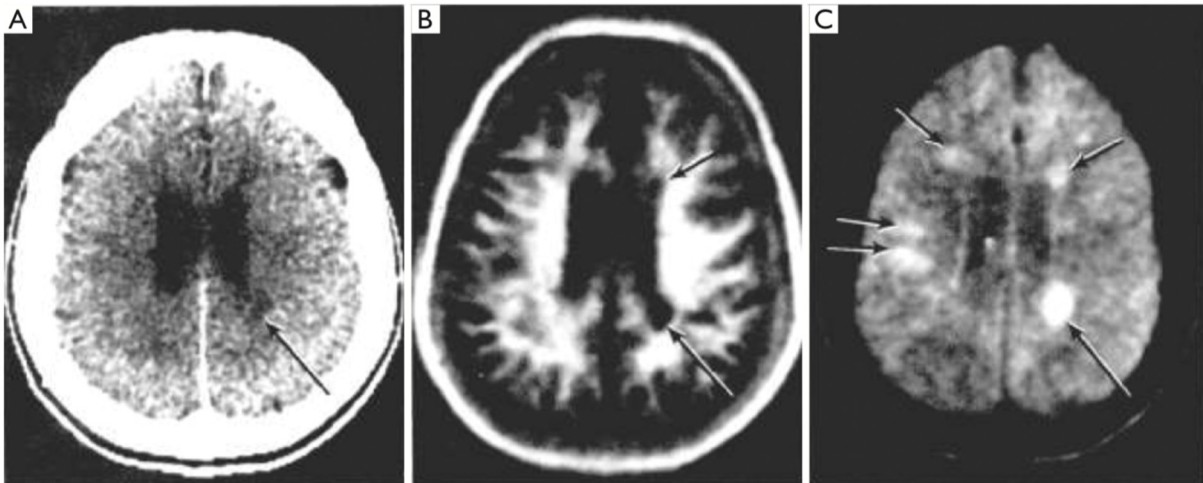

**Figure 8.** Multiple sclerosis (MS): CT (**A**), inversion recovery (IR) (**B**) and long TR, long TE spin echo (SE) (**C**) (July 1982) [8]. A single MS lesion is seen on the CT image (arrow) (**A**). This is also seen on the IR and SE images (arrows). There are an additional four lesions seen on the SE image (**C**) in normal appearing white matter on the CT image (**A**).

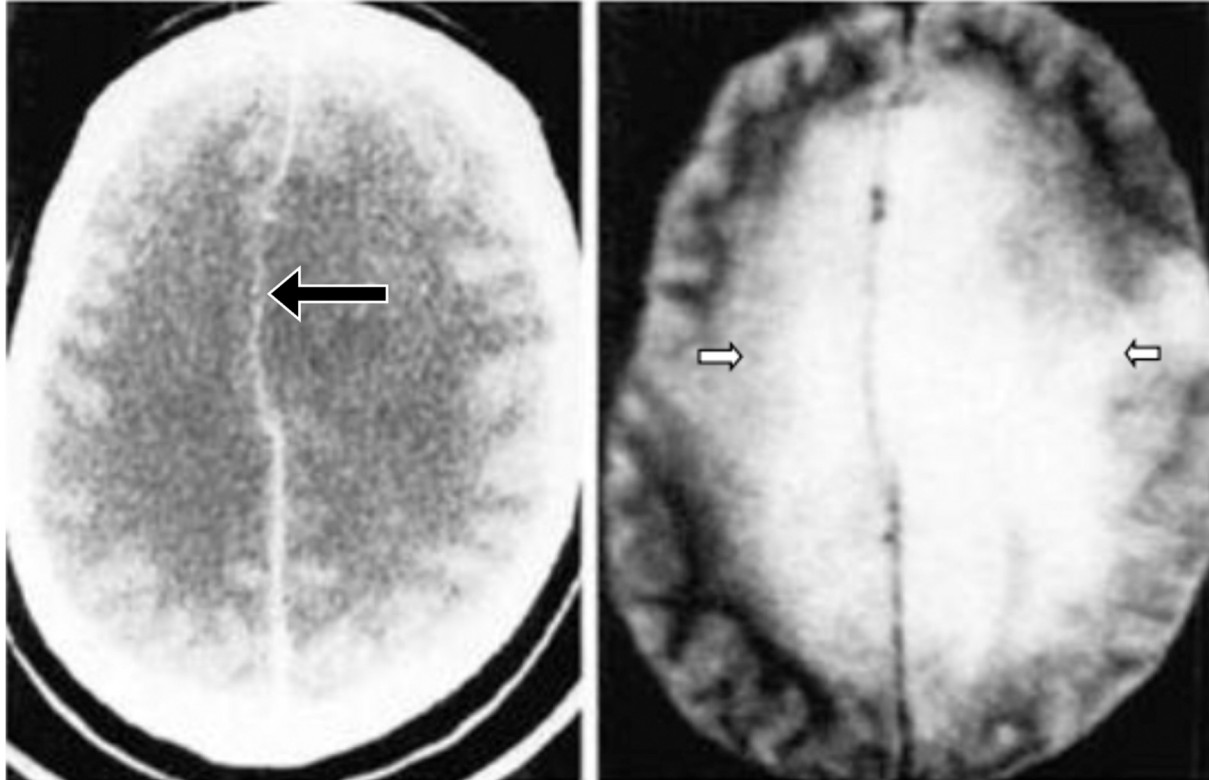

**Figure 9.** CT (**left**) and STIR (**right**) images of a glioma (August 1985) [10]. Displacement of the falx is seen on the CT scan (**left**, black arrow) but there are no other abnormal features. The STIR image (**right**) shows very extensive high signal abnormalities in both cerebral hemispheres (open arrows). The high signal abnormalities on the STIR image are in normal appearing white matter on the CT image.

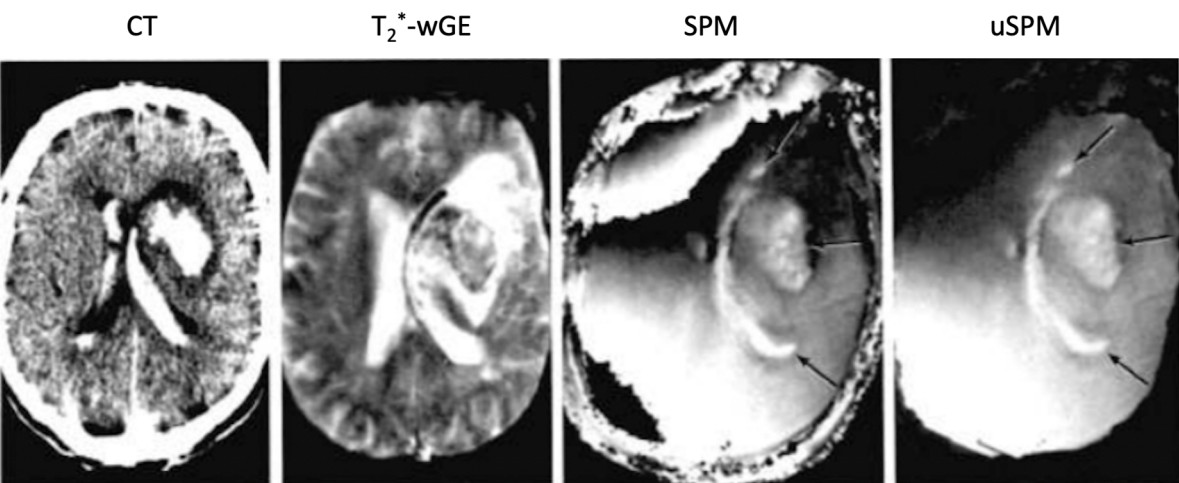

**Figure 10.** Intracranial hematomas: CT, heavily $T_2^*$-weighted gradient echo ($T_2^*$-wGE), susceptibility phase map (SPM) and partly unwrapped SPM (uSPM) 0.15 T (1987) images [12]. The CT image shows the intraparenchymal and ventricular hematomas with high signal. These images show some low signal on the $T_2^*$-wGE image. There are specific changes due to susceptibility contrast seen on the SPM and uSPM phase maps (arrows).

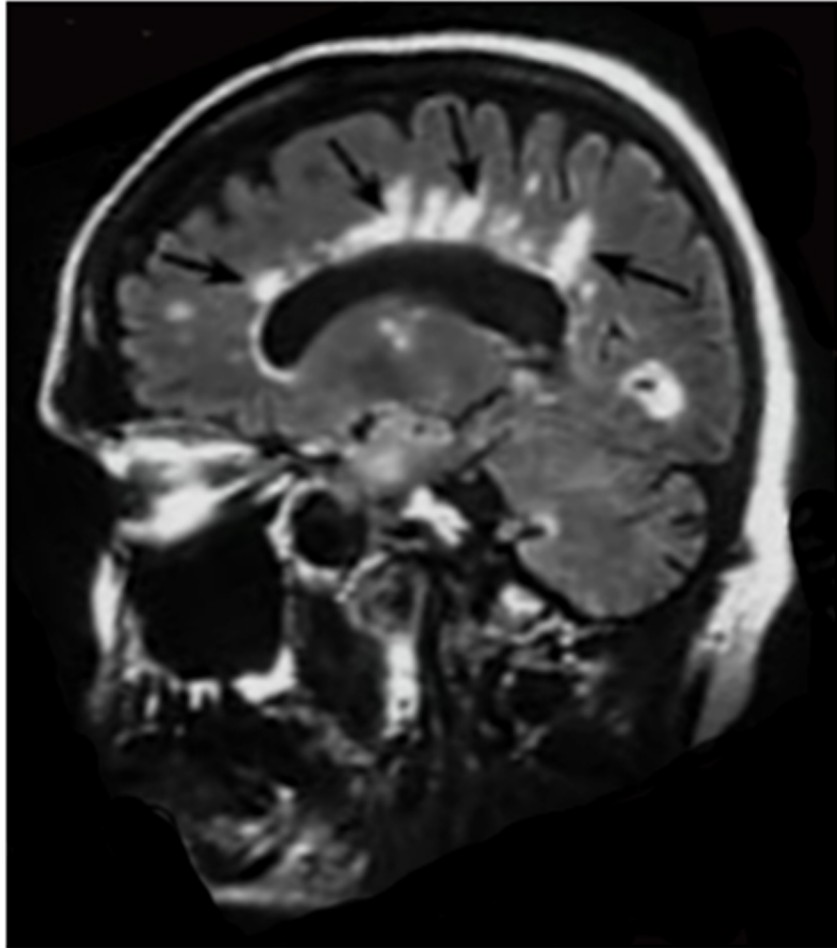

**Figure 11.** Multiple Sclerosis (MS): parasagittal $T_2$-FLAIR image. There are numerous MS lesions seen in the corpus callosum (black arrows) and around the occipital horn of the lateral ventricle. They are seen with high signal compared with the low signal of the brain and very low signal of the CSF.

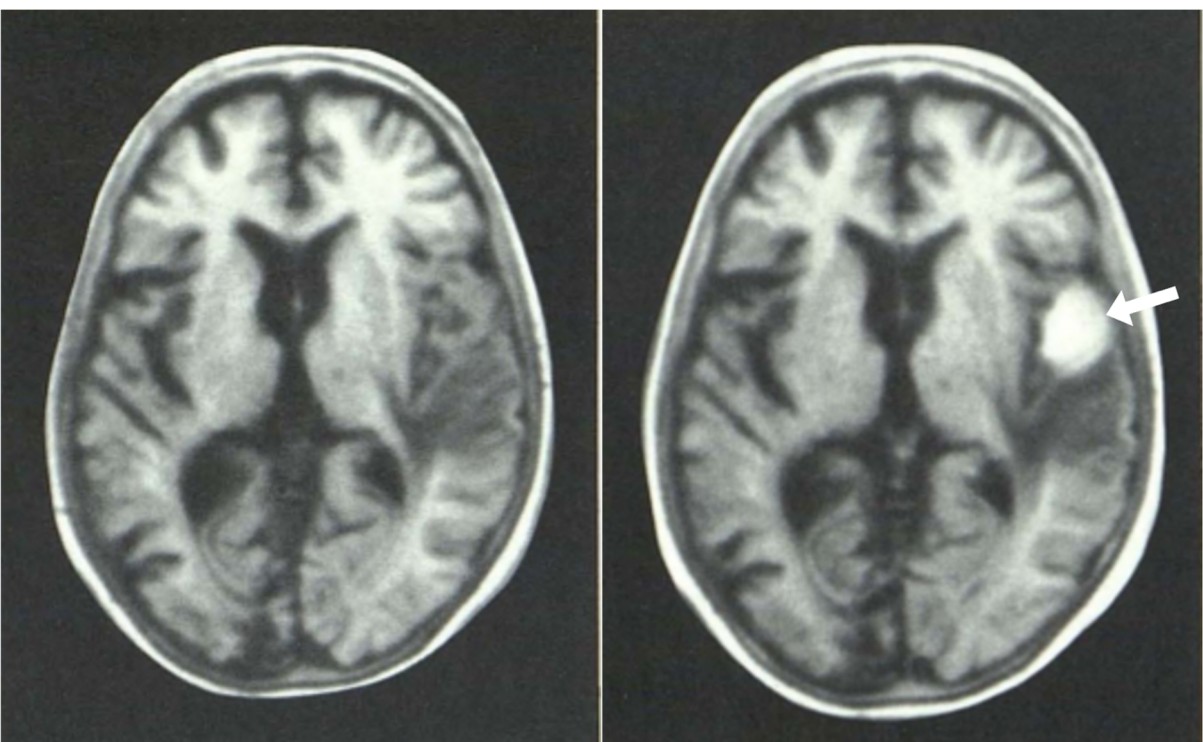

**Figure 12.** Glioma: pre (**left**) and post (**right**) intravenous GBCA shown with inversion recovery (IR) images (March 1984) [14]. The glioma and edema have a low signal on pre-enhancement (**left**) images but the signal in the tumor is increased by the GBCA and is seen with high positive contrast (white arrow).

However, there is evidence from postmortem studies that normal appearing tissues seen with MRI using present day protocols may actually be abnormal, and efforts have been made with techniques such as magnetization transfer and magnetic resonance spectroscopy (MRS) to demonstrate these abnormalities. These methods have not been successful enough for them to be included in most contemporary clinical imaging protocols.

At the present time, a third revolution in soft tissue contrast imaging is taking place in which normal appearing tissues seen with state-of-the-art MRI sequences show abnormalities with very high contrast when imaged with newer sequences such as divided subtracted inversion recovery (dSIR). In modelling studies, the dSIR sequence can show ten times more contrast than conventional IR sequences when imaging small changes in $T_1$ due to disease. The small changes in $T_1$ may be insufficient to generate contrast with conventional sequences but the much greater contrast amplification of the dSIR sequence can show obvious abnormalities.

The purpose of this paper is to describe the theory underlying the use of the dSIR sequence, and show how this sequence can make obvious a pattern of brain injury not previously described in MRI (the whiteout sign). This sign can be seen after different insults to the brain when little or no abnormality is apparent with conventional state-of-the-art sequences.

## 2. Theory

### 2.1. dSIR Sequences

In this section, the theory underlying the use of dSIR sequences is described. The description is included to provide a basis for interpretation of dSIR images as well as discussion of their features. More detailed descriptions of dSIR sequences and images have been published previously [16,17].

The mechanism underlying dSIR sequence contrast is shown in Figures 13 and 14.

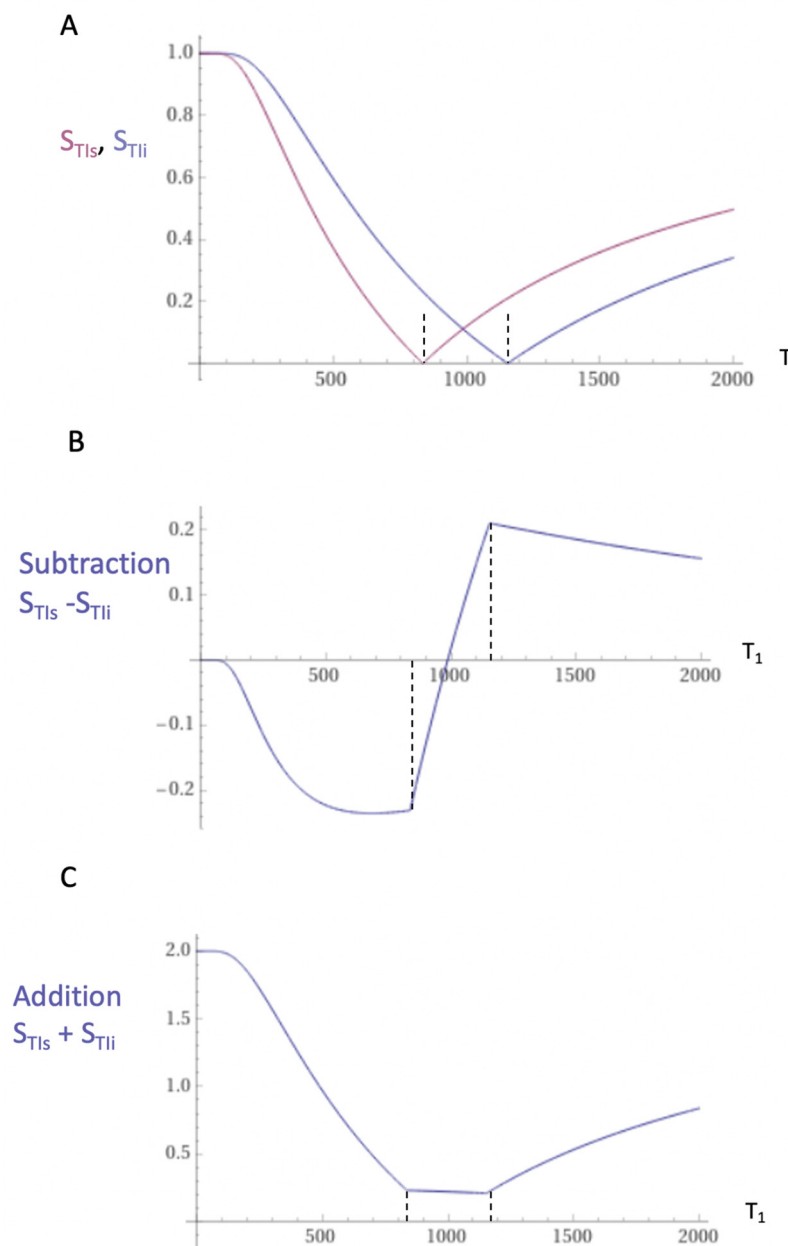

**Figure 13.** Subtracted IR (SIR) and Added IR (AIR) $T_1$-filters. $T_1$ is shown along the linear X axes in ms. (**A**) shows the $TI_s$ $T_1$-filter (pink) and $TI_i$ $T_1$-filter (blue), (**B**) shows the subtraction ($S_{TIs} - S_{TIi}$) IR or SIR $T_1$-filter, and (**C**) shows the addition ($S_{TIs} + S_{TIi}$) IR or AIR $T_1$-filter. The middle Domain (mD) is the $T_1$ values along the *X* axis between the vertical dashed lines. In (**B**), the slope of the SIR $T_1$-filter in the mD is approximately double that of the $S_{TIs}$ $T_1$-filter (pink in (**A**)). In (**C**), the signal at $T_1$ = 0 is doubled to 2.0, and the signal in the mD is reduced to approximately 0.20 in the nearly linear, slightly downward sloping central part of the AIR $T_1$-filter (i.e., in the middle Domain, mD).

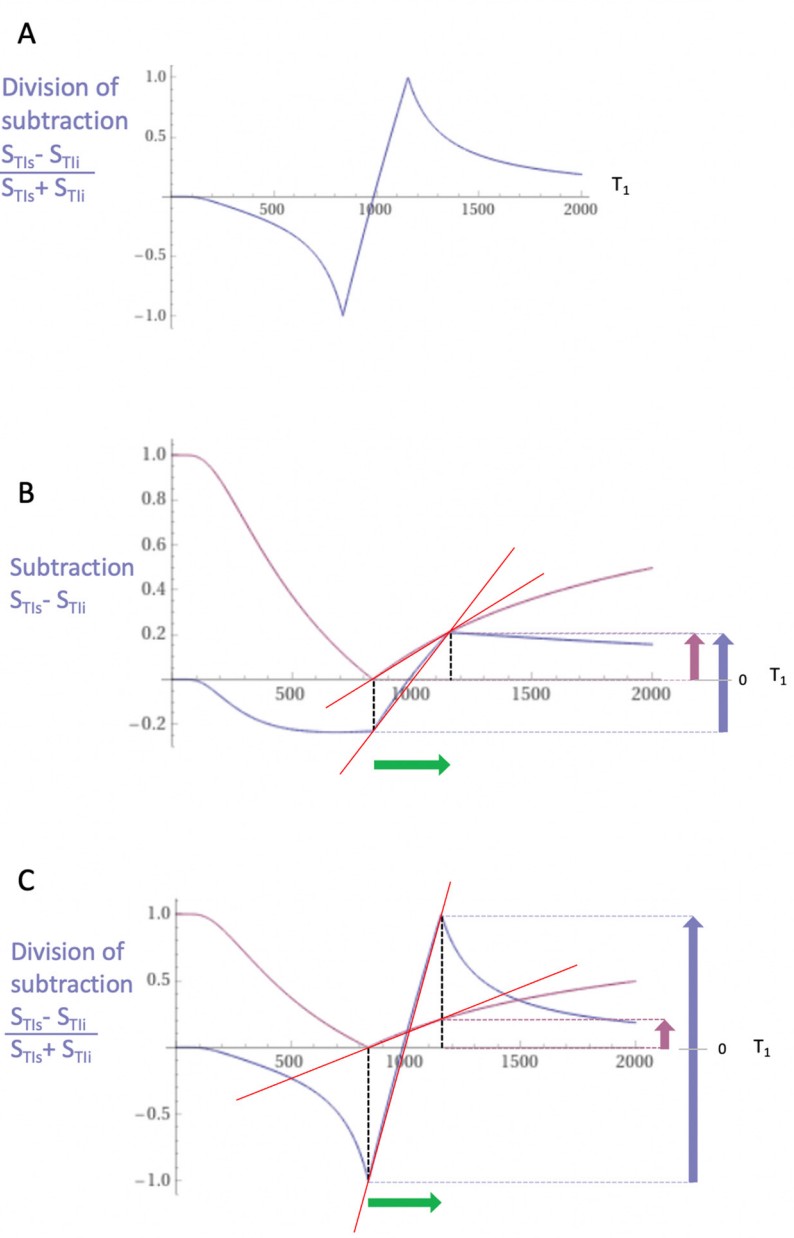

**Figure 14. Divided subtracted IR (A), subtracted IR showing contrast (B) and divided subtracted IR showing contrast (C) images.** (**A**) shows division of the SIR $T_1$-filter (**B**) by the addition $T_1$-filter (**C**) to give the dSIR $T_1$-bipolar filter. (**B**) shows comparison of the conventional IR $S_{TIs}$ $T_1$-filter (pink) and the SIR $T_1$-filter (blue) for a small increase in $T_1$ (horizontal green arrow, $\Delta T_1$). (**C**) is a comparison of the $S_{TIs}$ $T_1$-filter (pink) with dSIR $T_1$-filter (blue) for the small increase in $T_1$. In (**B**), the contrast produced by the SIR $T_1$-filter is twice that produced by the IR $T_1$-filter (blue and pink arrows). In (**C**), the contrast produced by the dSIR $T_1$-bipolar filter is ten times greater than that produced by the IR $T_1$-filter (blue and pink arrows).

Two magnitude reconstructed IR $T_1$-filters with different TIs: $TI_{short} = TI_s$ and $TI_{intermediate} = TI_i$ are shown in Figure 13A. $T_1$-filters are plots of signal against $T_1$ for MR sequences. When the second IR $T_1$-filter is subtracted from the first IR $T_1$-filter, this produces the Subtracted IR (SIR) $T_1$-filter in Figure 13B. In the central region or middle Domain (mD) of this SIR $T_1$-filter between the two vertical dashed lines shown in Figure 13B, the slope of the filter is approximately double that of the IR $T_1$-filters shown in Figure 13A.

The two $T_1$-filters shown in Figure 13A can be added to give the Added IR (AIR) $T_1$-filter as shown in Figure 13C. In its mD, which is bounded by the vertical dashed lines, the signal is reduced to approximately 0.10 of its value of 2 at $T_1 = 0$.

Figure 14A shows a dSIR $T_1$-bipolar filter in which the SIR $T_1$-filter in Figure 13B is divided by the AIR $T_1$-filter in Figure 13C. The resulting dSIR $T_1$-bipolar filter (Figure 14A) shows a highly positive nearly linear slope in its mD. Its slope is approximately ten times that of the IR $T_1$-filters shown in Figure 13A.

Figure 14B compares the contrast produced by the $S_{TIs}$ IR $T_1$-filter (pink) to that from the SIR $T_1$-filter (blue) from the same increase in $T_1$ ($\Delta T_1$) produced by disease (horizontal green arrow, $\Delta T_1$). Using the small change approximation of differential calculus, $\Delta T_1$ is multiplied by the slopes of the respective $S_{TI}$ IR and SIR $T_1$-filters (red lines) to produce the differences in signal $\Delta S$, i.e., contrast shown by the vertical pink and blue arrows on the right. The SIR $T_1$-filter generates approximately twice the contrast (blue arrow) of the $S_{TIs}$ dIR $T_1$-filter (pink arrow) from the same increase in $T_1$, $\Delta T_1$.

Figure 14C compares the contrast produced by the $S_{TIs}$ IR $T_1$-filter (pink) to that produced by the dSIR $T_1$-bipolar filter (blue). The increase in $T_1$ produced by disease (horizontal green arrow, $\Delta T_1$) is multiplied by the slopes of the respective $S_{TIs}$ IR and dSIR $T_1$-filters (red lines) to produce the differences in signal $\Delta S$, or contrast generated by the two $T_1$-filters. This is shown by the vertical pink and blue arrows on the right. For the same increase in $T_1$ ($\Delta T_1$), the dSIR $T_1$-bipolar filter produces approximately ten times greater contrast than the $S_{TIs}$ IR $T_1$-filter. The $S_{TIs}$ IR $T_1$-filter is that of a conventional IR sequence such as Magnetization-Prepared Rapid Acquisition Gradient Echo (MP-RAGE) sequence.

To produce the large increase in contrast shown in Figure 14C, dSIR sequences need to be correctly targeted at the normal tissue of interest as well as the small increase in the $T_1$ of the tissue produced by disease so that the change in $T_1$ (shown by the horizontal green arrow) is within the steeply sloping mD of the dSIR $T_1$-bipolar filter.

### 2.2. Contrast at Tissue Boundaries

At a boundary between two pure tissues (such as white and gray matter), the $T_1$s of voxels which contain mixtures of the two tissues within them typically span the range of $T_1$ values between those of the two pure tissues. If a narrow mD dSIR $T_1$-bipolar filter (e.g., with $TI_s$ nulling normal white matter, and $TI_i$ longer than $TI_s$ but less than that needed to null gray matter) is used, a $T_1$ value between those of the two pure tissues can result in a high signal $S_{W,G}$, for a value of $T_1$ between those of white and gray matter as shown in Figure 15. This high signal is seen at the boundary between white and gray matter and is shown on subsequent images (e.g., Figure 16). High signal boundaries between white and gray matter are a specific feature of narrow mD dSIR images. High signal boundaries are also seen at the junction between CSF and white matter at ventricular boundaries.

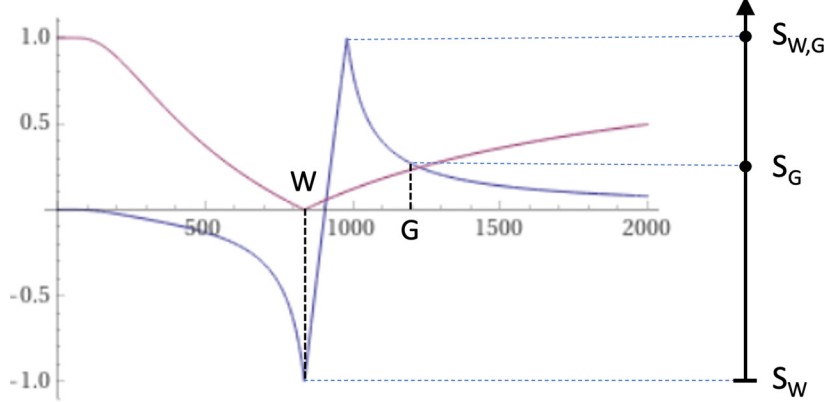

**Figure 15.** Boundaries. This image shows a narrow mD dSIR $T_1$-bipolar filter with a mD extending from white matter (W) to a $T_{1W,G}$ between the TIs of W and gray matter (G) (blue), and a white matter

nulled conventional IR $T_1$-filter, e.g., MP-RAGE (pink). The *X* axis is linear and is shown in ms. The peak signal ($S_{W,G}$) appears between W and G along the *X* axis, where there is a partial volume effect between W and G producing a $T_{1W,G}$ between the $T_1$s of W and G. This results in the high signal, $S_{W,G}$ between W and G. This corresponds to the high signal boundary between W and G shown in the following Figure 16 (left column).

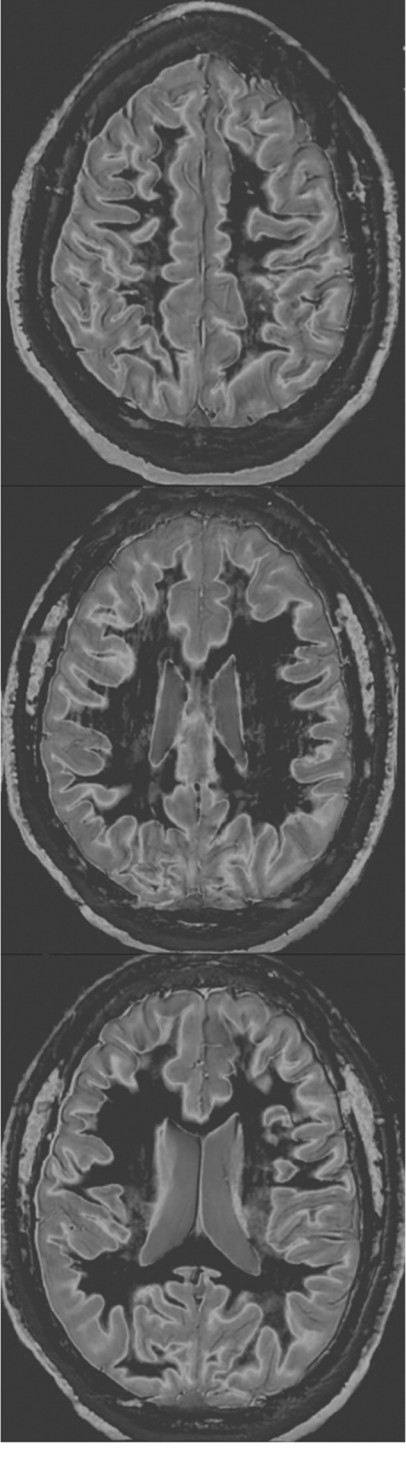

**Figure 16.** Normal 18-year-old control. 2D narrow mD dSIR images. The narrow mD dSIR images show normal white matter as very low signal intensity (dark) except for intermediate areas in and around the corticospinal tracts. This is a whiteout sign grade 1. Normal high signal boundaries are

seen at the junction between white matter and gray matter as well at the junction between white matter and CSF around the lateral ventricles.

### 2.3. $T_1$ Maps

To better understand the $T_1$-bipolar filter, a linear equation of the form $y = mx + c$ can be used to approximate the filter in the mD. The equation is produced by fitting a straight line between the first and last points of the mD (i.e., first point $x = TI_s/\ln 2$ and $y = -1$, and last point $x = TI_i/\ln 2$ and $y = +1$). In the mD, $S_{dSIR}$ is then given by:

$$S_{dSIR} \approx \frac{\ln 4}{\Delta TI} \; T_1 - \frac{\Sigma TI}{\Delta TI} \tag{1}$$

where $\Delta TI = TI_i - TI_s$ (i.e., second TI minus first TI), which is positive, and $\Sigma TI = TI_s + TI_i$. Note that because $\Delta TI$ is positive, the slope $\frac{\ln 4}{\Delta TI}$ is positive and the offset is negative.

The expression in Equation (1) captures four key features of the dSIR $T_1$-bipolar filter, firstly, the near linear change in signal (i.e., $S_{dSIR}$) with $T_1$ in the mD, secondly, the filter has a slope equal to $\ln 4/\Delta TI$, thirdly the filter shows high sensitivity to small changes in $T_1$ when the size of $\Delta TI$ is small. When $\Delta T_1$ is small, the size of $\Delta TI$ can be decreased to scale up the sensitivity. Such a reduction in $\Delta TI$ increases the steepness of the $T_1$-filter in the mD and the amplification of contrast is increased. As a result, contrast is maintained as $\Delta T_1$ and $\Delta TI$ decrease until the image becomes noise and/or artefact limited. This is not the case with conventional sequences when, if $\Delta T_1$ is decreased, contrast decreases.

Fourthly, Equation (1) can be used to map $T_1$ in the mD since:

$$T_1 \approx \frac{\Delta TI}{\ln 4} \; S_{dSIR} + \frac{\Sigma TI}{\ln 4} \tag{2}$$

The linear approximation is only valid in the mD. Also, it is assumed that TR is long compared to tissue $T_1$ values. An equivalent expression that allows for incomplete recovery of longitudinal magnetization during TR can be formulated.

Outside the mD in the highest Domain, the bipolar response may be approximated as the inverse of Equation (1).

$$S_{dSIR} \approx 1/S_{dSIR} \tag{3}$$

This captures another essential feature of the bipolar filter approach, i.e., reduction in out-of-band (i.e., out of mD) signals that would otherwise saturate the dynamic range. This characteristic provides broad contrast between the high dynamic range mD (black to white) against a lower contrast background of signals derived from shorter and longer $T_1$s outside the mD. This preserves anatomical coherence outside of the mD. Conventional window width narrowing coalesces out-of-band signals into solid blocks of black and white signal, and anatomical coherence is lost in these regions.

The high signal white boundary separating the mD from the highest Domain provides an easily identifiable boundary between tissue interfaces, as the $T_1$ of the voxels passes from those of the middle Domain to those of the highest Domain.

## 3. Methods

With approval from the New Zealand Health and Disability Ethics Committee (EXP 11360, 2022), and informed consent from each subject, MRI scans were performed on a 50-year-old male normal control, an 18-year-old male normal control, a 51-year-old male patient with methamphetamine use disorder and an 18-year-old male patient after a mild traumatic brain injury (mTBI). A 3T scanner (General Electric Healthcare, Chicago, IL, USA) was used. The 2D IR fast spin echo (FSE) sequences were performed with a $TI_s$ chosen to null a $T_1$ slightly shorter than the shortest $T_1$ of normal white matter, and a longer $TI_i$ chosen to produce narrow mD dSIR images targeted at small increases in the $T_1$ of white matter from normal as illustrated in Figure 14C. The initial $\Delta TI$ is slightly shorter than that required to null the shortest $T_1$ white matter (e.g., central anterior corpus

callosum) and was designed to accommodate variation in $T_1$. Positionally matched $T_2$-FLAIR images as described in Table 1 were also acquired.

**Table 1.** Pulse sequences and pulse sequence parameters used at 3T. Z = zipped.

| # | Sequence | TR (ms) | TI (ms) | TE (ms) | Matrix Size<br>Voxel Sizes (mm) | Number of<br>Slices | Slice Thickness<br>(mm) |
|---|----------|---------|---------|---------|-------------------------------|--------------------|------------------------|
| 1 | 2D FSE IR (for white matter nulling) | 9192 | 350 | 7 | 256 × 224<br>0.9 × 0.1<br>Z512<br>0.4 × 0.4 | 26 | 4 |
| 2 | 2D FSE IR (used with #1 for narrow mD dSIR) | 5796 | 500 | 7 | 256 × 224<br>0.9 × 0.1<br>Z512<br>0.4 × 0.4 | 26 | 4 |
| 3 | 2D $T_2$-FLAIR | 6300 | 1851 | 102 | 320 × 240<br>0.7 × 0.7<br>Z512 | 26 | 4 |

## 4. Results

### 4.1. Normal Control

MRI Findings

Figure 16 shows narrow mD dSIR images from the normal 18-year-old control. His white matter is normal and shows a low signal (dark) appearance with a mid-gray appearance in and around the corticospinal tracts (whiteout sign grade 1, Table 2). Normal high signal boundaries (white lines) are seen at junctions between white and gray matter.

**Table 2.** Grading system for the whiteout sign seen on dSIR images.

| Grade | Description |
|-------|-------------|
| 1 | Dark white matter of different degrees except for areas of normal mid gray in the corticospinal tracts and superior longitudinal fasciculi. The centrum semiovale is generally less dark. Cerebellar hemisphere white matter may appear dark gray and is more black than white and may be asymmetric (e.g., Figures 16 and 17, left column). |
| 2 | Small amount of increased signal in white matter. There should be clearly more black than white. Cerebellar hemisphere white matter may have mildly higher signal but is more black than white (Figure 19, right column). |
| 3 | Mild, generalized change, where it may be difficult to determine if there is more black than white or vice versa. There should not be more black than white signal; if this is so then the case is grade 2. At the level of the mid lateral ventricles, there are relatively large areas of peripheral white matter sparing. If there is little to no peripheral white matter sparing, the case is grade 4, not grade 3. |
| 4 | Nearly complete whiteout. Generally uniform high signal in the white matter, but not as high as with grade 5 and not as high as the boundary between white and gray matter. There can be small areas of peripheral sparing at the level of the lateral ventricles. Cerebellar hemisphere white matter is more white than black (e.g., Figures 17 and 18, right column, and Figure 19, left column). |
| 5 | Full whiteout. Nearly uniform high signal, nearly as high, or as high as the boundary between white and gray matter (it will not be higher). Little to no |

> low signal in the white matter of the centrum semiovale. Mild peripheral white matter sparing at the level of the lateral ventricles may be seen. The cerebellar hemisphere white matter may be high signal.

### 4.2. *Case 1 Methamphetamine Use Disorder*

#### 4.2.1. Case History

Case 1 was a 51-year-old male with longstanding methamphetamine use disorder. At eight years of age, he had a significant traumatic brain injury and was unconscious for several minutes following which he had a fall-off in educational function. He developed post-traumatic epilepsy, which was well controlled with sodium valproate initially, and more recently with levetiracetam. There were a large number of further brain injuries from falls, assaults and fights, none of which required surgical intervention. At some time, not accurately discernible now, he began to use methamphetamine. He continued to use this several times a week for the next 30 years. Many attempts were made to manage his methamphetamine use disorder over the years, with none of the standard abstinence models proving effective. This led more recently to him following a substitution model with methylphenidate.

On initial assessment, he presented in a way that focused on maintaining his substance use lifestyle. He was largely incapable of reflective thought processes and was obviously duplicitous in presentation of data. There were metacognitive issues with his being incapable of his recognizing that his duplicity was obvious to others. This added to issues with theory of mind making it easy for others to take advantage of him. Thought processes were largely concrete. Attentional function was poor with issues of attentional load (creating problems when stimulus load was increased), dividing and alternating attention and sustaining attention. He struggled to shift mental set, jumping from topic to topic due to his attentional issues.

After treatment began, he abstained from methamphetamine use and showed gradual cognitive and behavioral improvement. He thought in a rational and reflective manner with recognition of his own function and the impact he had on others. He projected into the future with plans that made sense. He remained habituated to some aspects of his lifestyle but with occupational therapy input he slowly improved. He underwent brain MRI scanning one month and nine months after methamphetamine abstinence while being treated with methylphenidate.

#### 4.2.2. MRI Findings

Figure 17, left column shows narrow mD dSIR images from the 50-year-old normal control (left column) and from the patient at his initial MR examination after one month's abstinence. The normal control shows a very low signal (dark) appearance in most of his white matter. There is an intermediate signal and higher signal in and around the corticospinal tracts (whiteout sign grade 1). In the right column, the patient's white matter shows extensive areas of high signal with only small areas of more normal lower signal white matter present in the periphery (white arrows) (whiteout sign grade 4).

There is a dramatic difference between normal white matter on the left which is black and abnormal white matter on the right which is white (apart from small spared or relatively spared areas).

Figure 18 compares $T_2$-FLAIR images (left column) with matched narrow mD dSIR images (right column) in Case 1 at the time of his first examination. No abnormality is seen in the white matter on the $T_2$-FLAIR images (left column) but extensive high signal abnormalities are seen in white matter on the narrow mD dSIR images. Small areas of more normal white matter with a low signal appearance are shown by the white arrows (right column). These are features of a whiteout sign grade 4.

Figure 19 shows narrow mD dSIR images in the patient after one month's abstinence (left column) and after nine months' abstinence (right column). Extensive abnormal high

signal is seen in white matter after one month's abstinence (left column). There is marked reduction in the extent of these high signal abnormalities in white matter after nine months abstinence from whiteout sign grade 4 to grade 2 (right column). The patient clinically improved during this period.

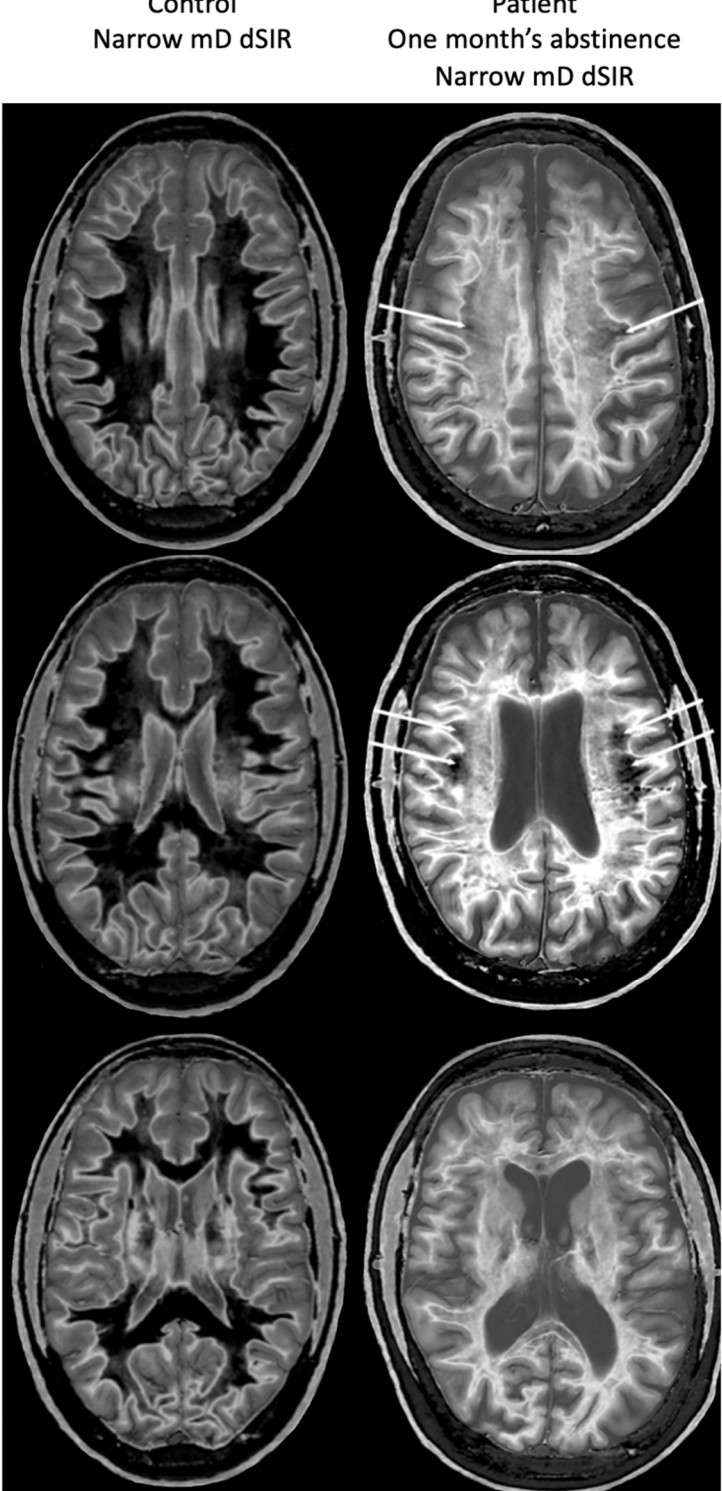

**Figure 17.** A 50-year-old normal control (**left column**) and Case 1, a 51-year-old with methamphetamine use disorder (**right column**). 2D dSIR narrow mD images are shown. The narrow mD dSIR images in the normal control show most white matter as very low signal intensity (dark) with a mid gray and lighter appearance in and around the corticospinal tracts, i.e., whiteout sign grade 1. The

narrow mD dSIR images in the patient show widespread high signal changes in white matter with only small areas of normal dark white matter at the periphery of the white matter (white arrows). The features are consistent with a whiteout sign grade 4. Normal high signal boundaries are seen between white matter and gray matter on the dSIR images in the control and the patient, but are less obvious in the patient because of the high signal in his abnormal white matter.

**Figure 18.** Case 1 patient with methamphetamine use disorder. Comparison of T2-FLAIR and narrow mD dSIR images. No abnormality is seen on the T2-FLAIR images (**left column**) but there are extensive areas of higher signal in approximately 90% of the white matter (**right column**). Only small areas of normal low signal are seen in the white matter (white arrows) (**right column**). The appearances on the dSIR images are consistent with a whiteout sign grade 4.

**Figure 19.** Case 1 patient with methamphetamine use disorder. Comparison of narrow mD dSIR images after one month's abstinence (**left column**) and after nine months' abstinence (**right**

**column**). After one month's abstinence the images show widespread abnormal areas of increased signal in white matter (**left column**). After nine months abstinence, the images show extensive dark low signal areas (**right column**) consistent with marked regression of his disease during the period between one and nine months' abstinence from whiteout sign grade 4 (**right column**) to grade 2 (**left column**). The boundaries between white and gray matter as well as between CSF and white matter around the lateral ventricles become more obvious after regression of the whiteout sign.

### 4.3. *Case 2 Mild TBI*

### 4.3.1. Case History

Case 2 was an 18-year-old male who clashed heads with another player while engaging in a variation in handball. There was no reported loss of consciousness or post-traumatic amnesia and he did not present to a healthcare professional for assessment of his injury. Case 2 was first scanned 21 h post-injury. At this time, he reported fatigue and attentional difficulties. A second scan was performed 64 h post-injury. At this later time, he did not report any mTBI-related symptoms.

### 4.3.2. MRI Findings

At his first examination 21 h post-injury, his $T_2$-FLAIR images were normal (Figure 20, left column) but his narrow mD dSIR images (right column) showed extensive high signal abnormalities in the white matter of his cerebral hemispheres with sparing of the anterior and posterior corpus callosum, as well as some sparing of the peripheral white matter in the cerebral hemispheres, consistent with a whiteout sign grade 4. This is quite different from the low signal seen in the white matter of the age matched normal control dSIR image shown in Figure 16, which is whiteout sign grade 1.

Figure 21 shows his dSIR images at 21 h post-injury (left column) and at 64 h post-injury (right column). There is a marked difference between the dSIR images at 21 h (left column) and 64 h (right column) consistent with complete regression of his whiteout sign over 43 h (i.e., in just less than two days) from grade 4 to grade 1. Case 2's 64 h post-injury dSIR images (right column) are similar to the normal control images shown in Figure 16.

No abnormality was seen on Case 2's $T_2$-FLAIR images either at 21 h or 64 h post-injury.

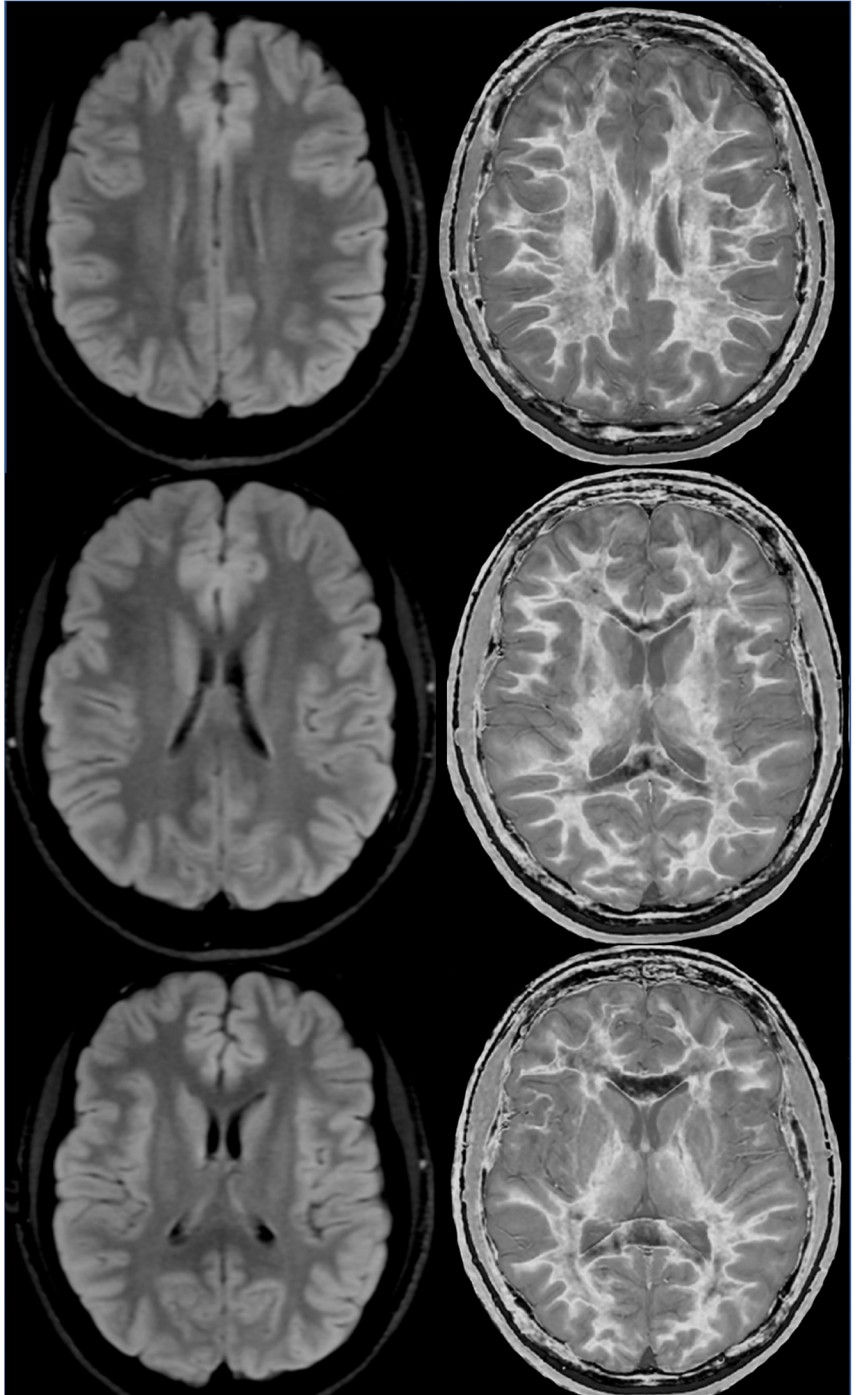

**Figure 20.** Case 2 Acute mTBI imaged at 21 h post-injury. T₂-FLAIR (**left column**) and narrow mD dSIR (**right column**) images. No abnormality is seen on the T₂-FLAIR images. On the dSIR images, extensive high signal abnormalities are seen in most of the white matter with sparing of the anterior and posterior central corpus callosum and posterior white matter of the cerebral hemispheres. The appearances are consistent with a whiteout sign grade 4.

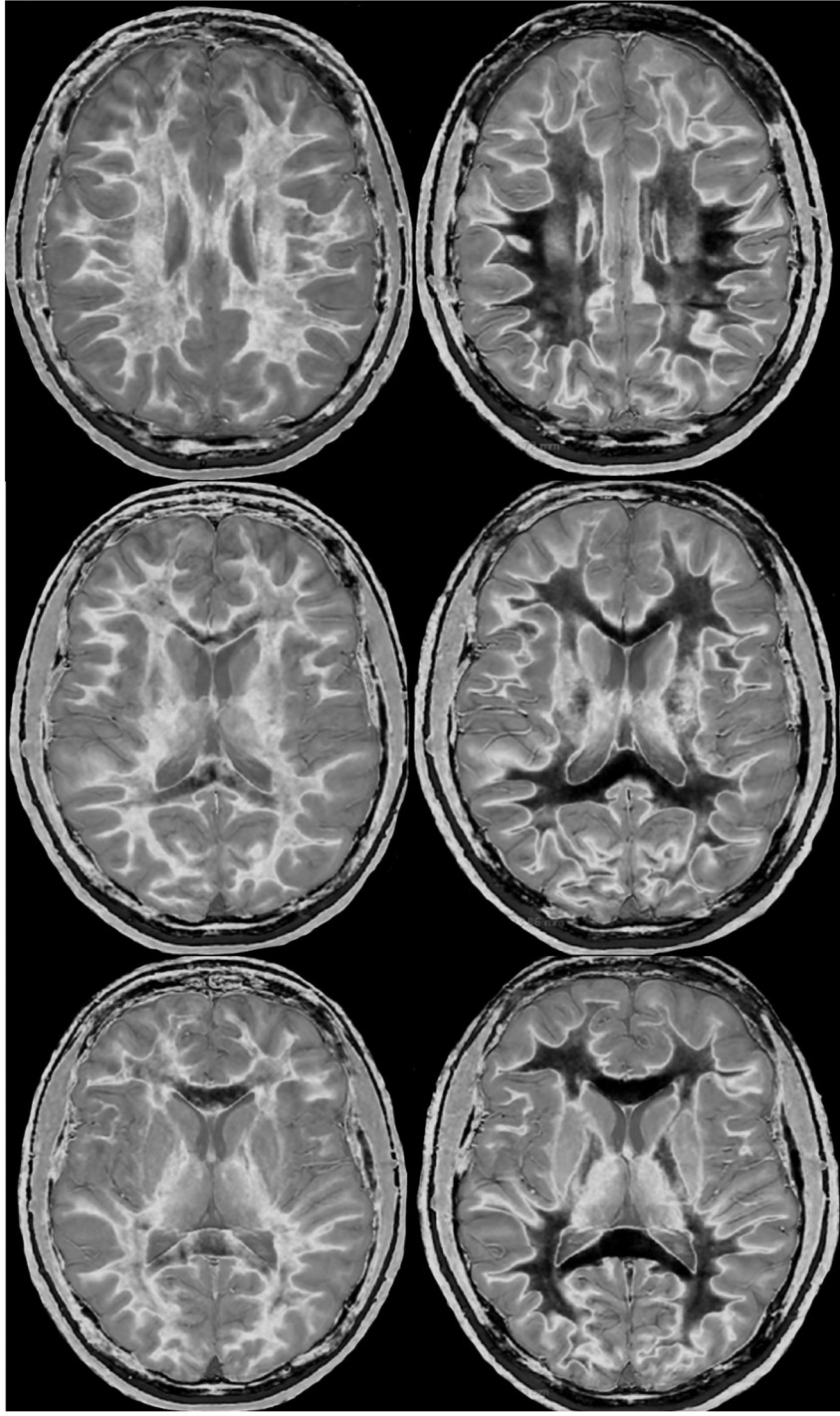

**Figure 21.** Case 2 Acute mTBI imaged 21 h and 64 h post-injury. Narrow mD dSIR images. On the images obtained at 21 h post-injury, extensive high signal abnormalities are seen in most of the white matter with sparing of the anterior and posterior central corpus callosum and peripheral white matter of the cerebral hemispheres (**left column**). On the images at 64 h post-injury, normal low signal white matter is seen in the hemispheres (**right column**). The images show marked regression of abnormalities over 43 h (i.e., just less than two days). The boundaries between white and gray matter and around the lateral ventricles become more obvious after regression of whiteout sign from grade 4 (**left column**) to grade 1 (**right column**).

## 5. Discussion

### 5.1. Ultra-High Contrast MRI

In general terms, it is possible to describe the soft tissue contrast produced by CT as high in relation to the soft tissue contrast seen on plain radiographs. The increased soft tissue contrast of MRI can then be described as very high, and that of recently developed MRI sequences such as dSIR as ultra-high (Table 3). This follows the naming of the radiofrequency spectral bands as high frequency (3–30 MHz), very high frequency (30–300 MHz) and ultra-high frequency (300–3000 MHz). Since the speed of radiowaves is $3 \times 10^8$ m/s, the radiofrequency bands can also be described in terms of wavelengths, i.e., high frequency (100 m–10 m), very high frequency (10 m–1 m) and ultra-high frequency (1 m to 1 mm).

**Table 3.** Comparison of divisions of imaging soft tissue contrast, the radiowave spectrum and static magnetic field strengths used in MRI.

| Concept | Descriptor | | |
|---|---|---|---|
| | *first level* | *second level* | *third level* |
| Soft tissue contrast | High | Very high | Ultra-high |
| | CT | Conventional MRI | dSIR sequences, etc. |
| Radiowave spectrum | High | Very high | Ultra-high |
| | 3–30 MHz | 30–300 MHz | 300–3000 MHz |
| Static field strength | Low | High | Ultra-high |
| | <1T | 1–7 T | 7–11.7 T possible 14 T |

This division also generally follows the classification of static field strengths used in clinical MRI which at the present time is low (less than 1 T with ultra-low < 0.1T), high (1–3 T) and ultra-high (7–11.7 T now, and possibly 14 T in the future). At 7 T, the proton resonance frequency is 300 MHz, which is approximately the operating frequency of a 7 T MRI system. As a result, the boundary between high frequency and ultra-high frequency radiowaves is essentially the same as that between high field and ultra-high field strength MRI systems.

The transition in contrast from high to very high contrast (CT to MRI) and from very high to ultra-high contrast (conventional MRI to dSIR and other forms of MRI) is typically manifest as normal appearing tissue seen with the first imaging modality showing unmistakable high contrast abnormalities with the second imaging modality. This is shown for the transition from conventional MRI to dSIR imaging in Figures 18 and 20, where no abnormality is seen in white matter on the $T_2$-FLAIR images, but extensive high signal changes are seen in white matter on the dSIR images.

A principal mechanism for increases in intrinsic contrast in MRI is synergistic contrast, in which: (i) a single tissue property such as $T_1$ is used twice or more in the same sequence to increase contrast, (ii) two or more different tissue properties such as $T_1$ and $T_2$ are used together to increase contrast, and (iii) both techniques (i) and (ii) are used in a single sequence.

An example of the use of $T_1$ and $T_2$ to create synergistic contrast is the STIR sequence, where increases in both $T_1$ and $T_2$ result in additive contrast. Another example is diffusion-weighted (D*-weighted) imaging with a long TE sequence. For an increase in $T_2$ and a decrease in D*, the $T_2$ and D* contrast is synergistic and overall contrast is increased. If, as is common, both $T_2$ and D* are increased in disease then the contrast produced by each of these tissue properties is opposed and the overall result is a reduction in contrast, which is not usually clinically helpful.

With a single tissue property such as $T_1$, it is possible to approximately double the contrast produced by small changes in $T_1$ compared to a conventional IR sequence using subtraction of one image from the other (Figure 13). However, more dramatic improvement in contrast comes with division of the subtracted image by the sum of the two images

(Figure 14). With this approach, the increase in contrast relative to the conventional IR sequences may be ten or more times. Normalization by the local signal intensity removes coil shading. Mobile proton density and $T_2$-weightings are also removed by normalization leaving only magnified $T_1$ contrast. The size of the increase in contrast is eventually limited by signal-to-noise (SNR) and artefact constraints.

The effect of GBCAs in shortening $T_1$ can also be amplified using divided reverse subtracted (drSIR) sequences. This allows ultra-high contrast to be generated by small reductions in $T_1$ produced by the GBCAs which are insufficient to produce contrast with conventional $T_1$-weighted IR sequences.

dSIR sequences can usually be implemented on existing MR machines using IR pulse sequences that already exist on MRI systems. Ultra-high contrast can often be achieved at the same spatial resolution as conventional techniques. This may require some increase in time but typical dSIR acquisitions can readily be performed in less than five minutes on most MRI systems. The software required for the IR image manipulation necessary to produce dSIR sequences only involves basic arithmetic and is easily written in MATLAB (Natick, MA, USA).

*5.2. The Whiteout Sign*

The whiteout sign shows a bilateral, symmetrical and generally uniform increase in signal in white matter of the cerebral and cerebellar hemispheres. This increase in signal is relative to the normal signals seen in white matter. The normal anterior and posterior central corpus callosum has a very low signal. The corticospinal tracts and their surrounds, as well as the superior longitudinal fasciculi have higher signal than white matter elsewhere in the hemispheres. The increase in signal with the whiteout sign is superimposed on this normal pattern.

The whiteout sign usually shows sparing or relative sparing of the anterior and posterior central corpus callosum (i.e., the genu and splenum of the corpus callosum) as well as adjacent white matter (forceps minor and forceps major). There is usually also sparing or relative sparing of the peripheral white matter of the cerebral hemispheres.

The recovery or reversal phase of the whiteout sign (i.e., return towards normal) proceeds from the anterior and posterior central corpus callosum outwards, the periphery of the cerebral hemisphere white matter inwards, and the central regions of the cerebellar hemisphere white matter outwards.

There may be small focal or multifocal high signal changes present with other features of the whiteout sign. These are made more obvious by having low signal (more normal) white matter around them in, for example, the peripheral white matter of the cerebral hemispheres.

There are differences in the degree of the whiteout sign both in the increase in signal and the extent of the changes within white matter. Qualitative assessment may be by both by the degree of increase in the signal and the extent of the abnormality as a proportion of the total white matter seen in a slice in the cerebral or cerebellar hemispheres.

A grading system can be used to describe these changes (Table 2). The system used to date has five grades 1 (normal) to 5 (maximum abnormal). The normal grade is low signal (dark appearance). The central corpus callosum and peripheral white matter in the cerebral hemispheres are usually lowest signal within the generally low signal pattern. The centrum semiovale is slightly higher signal as are the cerebellar hemispheres (Table 3). The maximum abnormal grade is characterized by high signal (light appearance) with partial or complete loss of the high signal boundary between white and gray matter. Use of the grading system is illustrated in Figures 16–21.

In preliminary studies, a cutoff point between grades 1 and 2 as normal and grades 3,4,5 as abnormal correlated very closely with clinical assessment of cognitive impairment in patients with mTBI and appears likely to be useful in clinical practice.

Quantitation of $T_1$. dSIR images are $T_1$ maps and in the mDs signal is nearly linearly proportional to $T_1$ (Equation (2)). It is therefore possible to calibrate the display gray scale

in terms of $T_1$. This has the advantages over conventional $T_1$ mapping in that it requires no extra acquisition, it is at the same spatial resolution as the clinical image (since it is the clinical image) not the lower spatial resolution generally used for $T_1$ maps, and no registration procedure is required to align the clinical images and the $T_1$ maps.

It is possible both to directly read values of $T_1$ and to measure differences in $T_1$ between regions on dSIR images and determine how these correlate with visible contrast. This applies both to normal and abnormal white matter. The values of $T_1$ do not apply outside of the mD, so values for gray matter are in error when white matter is targeted with dSIR sequences. Also, the values of $T_1$ assume full relaxation with a long TR. If TR is less than that needed to allow full relaxation of a tissue, or if a shorter duration TR is desired, a more general expression for $T_1$ recovery than those used in Equations (1) and (2) may be used to calculate suitable TIs. Quantitation of both increase in $T_1$ and the extent of the abnormality can be combined.

The whiteout sign may be seen in acute, recurrent and chronic disease with natural histories including complete reversal as well as persistence. It may be seen together with signs of other diseases such as infarction, hemorrhage and small vessel disease.

When the whiteout sign is seen, there is usually little or no abnormality present in the corresponding regions in the brain using conventional $T_2$-weighted images such as $T_2$-wFSE and $T_2$-FLAIR sequences. To date, the whiteout sign has only been shown with dSIR sequences and these are described in more detail in the next section.

### 5.3. dSIR Sequences

The dSIR sequence is one of the class of Multiplied Added Subtracted and/or Divided Inversion Recovery (MASDIR) sequences. It employs a subtraction of signal from a longer TI image from a shorter TI image. This is divided by the sum of the two images. As a consequence, the dSIR sequence is very largely independent of mobile proton density ($r_m$) and $T_2$. It is essentially a pure measure of $T_1$. The relationship between signal and $T_1$ is very closely linear within the mD. Thus, images can be regarded as quite accurate $T_1$ maps in the mD while outside the mD signals fade to mid-gray (zero) as $T_1$ is decreased or increased. The sequences can be used in 2D and 3D (isotropic or anisotropic) forms. The data collection may be FSE or gradient echo (GE), both usually with short TEs to increase SNR. The value of TE/TR is not critical since the $T_1$ information is determined by the difference in TI.

The dSIR sequence is used in targeted form. The first TI is typically chosen to null the shortest $T_1$ in the normal white matter of interest. The second TI is chosen to accommodate small increases in the $T_1$ in normal white matter and keep these within the mD as illustrated in Figure 14C. The sequence is highly sensitive to small increases in $T_1$ and produces much greater contrast from those changes in $T_1$ than conventional MP-RAGE sequences. For changes in $T_1$ outside the mD, the sensitivity to change is generally much less, and the contrast may "overshoot" and be reduced when there are increases in $T_1$ that take the change in $T_1$ outside of the mD.

There are high signal lines which have an "etched" appearance between white matter and gray matter as well as between white matter and CSF. These high signal lines show the extent of white matter in the cerebral and cerebellar hemispheres.

The boundaries arise because partial volume effects between two tissues (or a tissue and a fluid) with different $T_1$s in mixed voxels produce $T_1$s which correspond to the values of $T_1$ required to produce the maximum signal of the $T_1$-bipolar filters shown on Figures 14C and 15. If the $T_1$ and signal of white matter is increased, then the contrast between the abnormal white matter and the high signal may be reduced so the boundary appears less obvious.

It is possible to create synthetic dSIR images from $T_1$ maps using $T_1$-BipoLAr fIlteRs ($T_1$-BLAIRs). These synthetic dSIR images may have essentially any chosen values of TI and may be targeted at particular tissues and specific changes in the $T_1$ of these tissues in

disease. It is also possible to create synthetic narrower mD dSIR images from wider mD dSIR images.

Synthetic TP-bipolar filters may also be applied to other TP maps in addition to $T_1$ maps such as $T_2$, $T_2$*, D* and $\chi$ to create synthetic $T_2$-BLAIR, $T_2$*-BLAIR, D*-BLAIR and $\chi$-BLAIR images. Two or more synthetic TP-BLAIR images may also be multiplied together to create synthetic multi-TP contrast, e.g., $T_1$, $T_2$-BLAIR images. The term $T_1$-BLAIR can be used generally to include both directly acquired dSIR and drSIR images as well as synthetic versions of these images, both with $T_1$ as the tissue property. Synthetic TP-BLAIR with other TPs such as $T_2$, $T_2$* and D* are also included in the generic category of TP-BLAIR images.

### 5.4. Post-Insult Leukoencephalopathy Syndromes (PILS)

Post-insult leukoencephalopathy is the name given to syndromes in which, following an insult to the brain, white matter changes are seen in a whiteout sign as shown in Cases 1 and 2 in this paper.

The causes of the syndrome observed to date have been methamphetamine use disorder in this paper and reference [17], mTBI (acute in this paper, and recurrent in reference [18]) as well as Grinker's myelinopathy (delayed post-hypoxic leukoencephalopathy [19]. In spite of their disparate etiologies, a stereotypical pattern of change in the white matter has been seen (i.e., the whiteout sign) in these conditions [20]. Other possible causes of the syndrome include other drugs, e.g., opiates, and post-viral infections, e.g., long COVID.

The severity, forms of injury and time course may all be important in inducing the whiteout sign. There may also be individual differences in patient susceptibility to insults that may result in the whiteout sign.

The symptoms and signs of the PILS may vary with the cause and severity of the insult. These may also affect the time course of symptoms and signs. Cognitive impairment is a feature and its time course may parallel the onset and remission of the whiteout sign.

### 5.5. Pathophysiology and Pathology

The bilateral symmetrical features with mainly uniform signal favor generalized pathological processes as the origin of the whiteout sign. These processes include neuroinflammation, particularly in the acute phase, but also chronically. Demyelination and degeneration may also be important. The rapid reversibility in acute cases favors an edematous and/or inflammatory process. It is likely that more than one pathophysiological process is involved. Histological validation of the appearances of the whiteout sign is likely to require animal studies, as is the study of the evolution of the sign over time.

### 5.6. Normal Appearing White Matter

This term is applied to white matter which appears normal with conventional MRI sequences, but there may be suspicions that this white matter is actually abnormal. Methods of trying to demonstrate this include magnetization transfer, diffusion and proton spectroscopy, none of which have achieved an established role in routine clinical diagnosis of suspected white matter abnormalities. The dSIR approach differs in that changes in a tissue property used in routine clinical diagnosis, i.e., $T_1$ (as with MP-RAGE sequences) are used to demonstrate abnormalities in normal appearing white matter not a different tissue property.

### 5.7. Validation

In the absence of pathological verification, validation of the imaging findings is indirect and is based on:

- The theory is summarized in Figures 13 and 14. This provides a consistent account of the contrast seen on dSIR images.

- Normal controls which show low signal in a characteristic pattern in normal white matter with dSIR sequences as illustrated in the two normal controls included in this paper. Also, the patient can in effect act as her/his own control when the abnormal changes revert to normal or near normal as shown in the two cases included in this paper.
- Boundaries between white matter and gray matter, and between white matter and CSF. These are a distinct feature of dSIR images and can be explained using the model in Figure 15. They are always seen in normal subjects and patients when the dSIR sequence is correctly performed. Boundaries of this type have not been seen with any other sequence. This supports the validity of the model described using Figures 13 and 14.
- There is also consistency between lesions with large changes in $T_2$ (as well as $T_1$) seen on $T_2$-FLAIR images in the same location as changes on dSIR images. The features include high signal boundaries and the lesions on dSIR images as predicted by them.
- The similar appearances of whiteout signs despite the different insults causing them suggests that there is a common pathophysiological process underlying them.
- A phantom containing solutions with known $T_1$s was imaged with IR FSE sequences with TIs ranging from 24 to 1024 ms. Measurements of signal taken with dSIR sequences with TIs of 124 and 524 ms are shown (Figure 22). These values showed very close agreement with the theoretical dSIR curves shown in Figures 14C and 15 with correspondence to the values of $T_1$ of the solutions [21]. These quantitative results support the validity of the numerical simulations shown in Figures 13 and 14.
- There has been a similarity of appearances on dSIR images obtained on GE, Philips and Siemens images at both 1.5 and 3.0 T, so the findings are not machine or field strength specific.

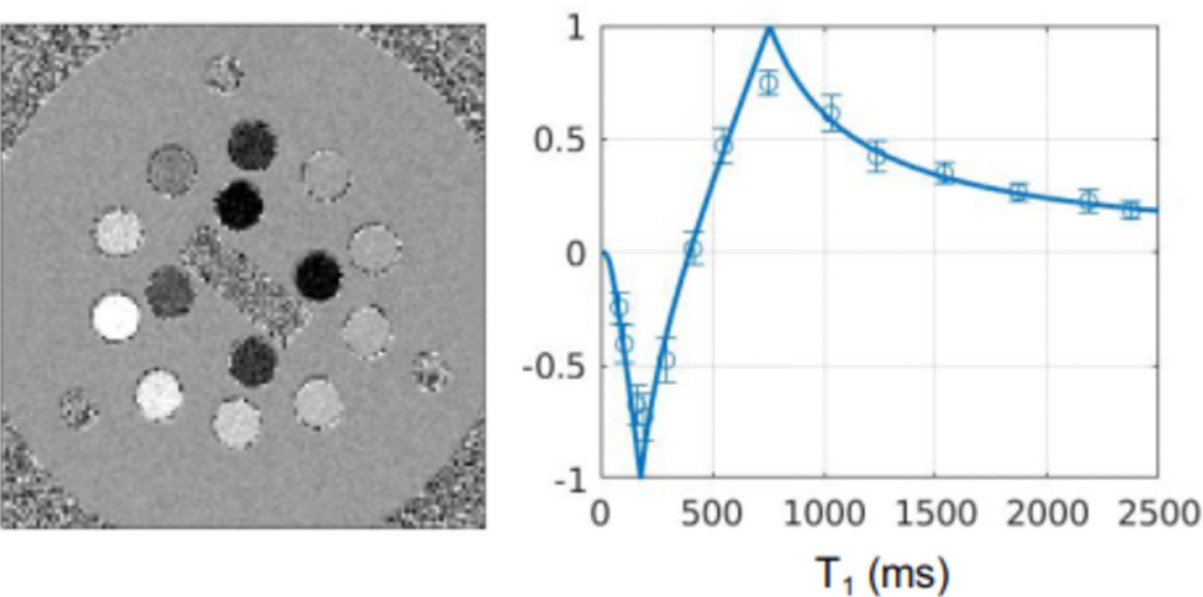

**Figure 22.** dSIR image with TIs 124 ms and 524 ms showing $T_1$-dependent signals in each compartment of the phantom (**left**) and plot of the measured signal (data points) in each compartment (**circles**) ± std versus known $T_1$ (**right**). The theoretical curve is overlaid on the data points (**right**). There is close agreement between the theoretical dSIR $T_1$-bipolar filter and the data points.

*5.8. Multiple Sclerosis (MS)*

In multiple sclerosis (MS), both focal features and generalized abnormalities have been observed [16] but, to date, these do not usually have the bilateral symmetry and generally uniform signal with sparing in a specific pattern that is seen with the whiteout sign.

It is possible that MS may have been precipitated by an immune response following an initial Epstein–Barr viral infection. However, significant time may have elapsed between this event and the MR examination. During this time, there may also have been exacerbations and remissions at different times and in different parts of the brain breaking up the uniform pattern seen with the whiteout sign.

The focal and multifocal features are usually more prominent in MS and may follow other particular distributions, e.g., along cerebral veins in the form of Dawson's fingers. The lesions seen in MS tend to be focal, signal overshoot is common and apparent "outpouching" of the ventricular system is seen when the high signal boundary around the ventricular system is lost at the site of an adjacent MS lesion.

### 5.9. Diffuse Disease in White Matter Seen with Conventional MRI Sequences

The whiteout sign is usually not associated with abnormalities seen on $T_2$-wFSE or $T_2$-FLAIR, but there are situations where bilateral and often symmetrical changes may be seen with conventional sequences. The relation of these to dSIR imaging of the same patients is a topic of considerable interest.

Grinker's myelinopathy or delayed post-hypoxic leukoencephalopathy may produce extensive white matter changes that are obvious on $T_2$-wFSE and/or $T_2$-FLAIR images. The condition is thought to be rare. The whiteout sign seen in post-hypoxic patients on dSIR images may actually be a less severe form of the changes seen in Grinker's myelinopathy in which changes are not seen with $T_2$-wFSE and $T_2$-FLAIR images [19]. It is therefore possible that Grinker's myelinopathy is much more common than usually thought, but is not recognized because the changes in white matter are usually insufficient to produce diagnostic contrast with conventional sequences.

Posterior reversible encephalopathy syndrome (PRES) is a condition induced by a variety of different insults [22]. There are usually bilateral symmetrical changes in the posterior white matter of both hemispheres but changes may be seen elsewhere in the brain. dSIR imaging could show more extensive changes.

Diffusely abnormal white matter (DAWM) in MS. This is a condition in which, using conventional sequences, diffusely abnormal increase in signal is seen in up to 25% of cases of MS [23]. These patients may show a more generalized pattern of abnormality than the MS patients examined to date with dSIR sequences.

The leukodystrophies. These often show diffuse abnormal high signal in white matter. The distribution in metachromatic leukodystrophy is similar to that seen with the whiteout sign but these are not reversible. Normal appearing white matter in the leukodystrophies may show abnormalities with dSIR sequences.

### 5.10. Clinical Value

Showing the whiteout sign as evidence of brain disease, using dSIR sequences may be of importance in distinguishing organic and psychological origins of disease. The demonstration of brain changes may also be of prognostic value and importance in monitoring the effects of treatment in a variety of different conditions.

### 5.11. Amplified MRI (aMRI)

Amplified MRI in which a video of brain motion with arterial pulsation is recorded and the displacement is amplified ten or more times is another example of ultra-high contrast MRI [24]. The tissue motion is not apparent with conventional imaging, but is obvious with aMRI.

### 5.12. Advances in Ultra-High Contrast MRI

Short term technical developments include implementation of 3D forms of dSIR which are likely to require different TIs than the 2D dSIR sequence for this study. A 1 mm$^3$ or smaller isotropic 3D imaging will facilitate imaging of the cerebral cortex as well as the

central gray matter and brainstem. It will also facilitate GBCA studies and other serial examinations in which rigid body registration can be used.

There are also technical advances in AI, image noise reduction and registration which may improve the quality of dSIR images.

Additional imaging dSIR images can be created synthetically from $T_1$ maps. It is also possible to synthetically create narrower mD images from wider mD images. The $T_1$ maps may be produced by MR fingerprinting and other methods such as actual flip angle imaging which do not involve an IR sequence so that the more general term $T_1$-BipoLAr fIlteR ($T_1$-BLAIR) imaging may be preferred to describe imaging utilizing a synthetic $T_1$-bipolar filter. The term $T_1$-BLAIR includes direct acquisitions (such as with dSIR) as well as synthetic imaging utilizing $T_1$ maps generated in different ways.

### 5.13. Ultra-High Spatial Resolution MRI

David Feinberg has supervised the construction of a head only 7T systems which produces ten times the spatial resolution (i.e., voxel size 0.2 × 0.2 × 1 mm) of conventional 3T clinical MRI systems [25]. This NexGen system incorporates three layer gradients which have performance specifications of 200 mT/m gradient strength and 900 T/m/s slew rate. The cost to date has been USD 22 M. The system is particularly designed for high spatial resolution imaging of the cerebral cortex.

### 5.14. Ultra-High Field MRI

There are now an estimated 120 ultra-high field MRI systems in operation with the majority 7 T systems. This is approximately 0.2% of the 60,000 MRI systems installed Worldwide.

Notable installations at field strengths above 7 T are whole-body systems operating at 10.5 T [26,27] and 11.7 T [28]. The latter system cost approximately USD 75M. Planning is underway for systems operating at 14T in Germany [29] and the Netherlands [30]. SNR ratio increases linearly with field strength or better (e.g., to the power of 1.65) and this results in a potential gain of approximately an order of magnitude in SNR when the static field strength is increased from 3 T to 11.7 T. This can be used to increase the speed of scanning, spatial resolution and/or contrast.

### 5.15. Summary

Another revolution in imaging soft tissue contrast is now in progress. The first major revolution was from plain radiographs to CT, the second major revolution was from CT to conventional MRI and the current one is from conventional MRI to ultra-high contrast MRI. The revolutions are characterized by change from invisible lesions with one modality to lesions seen with high contrast with the other modality.

With dSIR the spatial resolution is essentially the same as conventional images and the acquisition times are similar. This is in order to make the technique clinically realizable. The increase in contrast may be ten times greater than that with conventional IR sequences. This increase in contrast is typically targeted at normal appearing tissues, where there are only small changes in $T_1$ and/or $T_2$ present and these are insufficient to produce useful contrast with conventional state-of-the-art sequences. Thus, dSIR sequences are targeted to produce amplified contrast where it is most needed, i.e., to provide clinically useful visualization of the abnormalities which are not otherwise seen. Contrast is not increased in areas where there is already high contrast available on conventional images due to large changes in $T_1$ and/or $T_2$. Contrast amplification is not needed in this situation.

The changes in $T_1$ seen with the whiteout sign are only small, but they are widespread and so high spatial resolution is not necessary to see them.

The whiteout sign may be part of a generalized neuroinflammatory response to insults to the brain of various types. It may be initially due to edema and acute inflammation and may regress over two days or less. It may also become persistent for years and its

character may change to include demyelination and degeneration. Although the changes in $T_1$ associated with the whiteout sign are small in size, they are widespread within white matter and this may result in them having significant clinical impact.

The dSIR technique is ideally used as complementary to $T_2$-FLAIR. The $T_2$-FLAIR sequence shows high contrast from larger changes in $T_2$ against a generally bland background and has the advantage of clarity in demonstrating abnormality in comparison with sequences such as $T_2$-wFSE and conventional IR sequences. The dSIR sequence typically shows abnormalities due to small changes in $T_1$ and these may be present in the normal appearing white matter seen on positionally matched $T_2$-FLAIR images. The sequences are thus complementary.

Ultra-high contrast dSIR sequences are easy to implement on existing MRI systems using sequences already on the systems, and the cost of this is a tiny fraction of that of building and installing ultra-high spatial resolution MRI or ultra-high field MRI systems. In addition, dSIR sequences can readily be implemented and successfully tested on volunteers in a day or two. Another major advantage of ultra-high contrast MRI is that it can, in principle, be implemented on MRI systems operating at any field strength. This is not the case with ultra-high spatial resolution MRI or ultra-high field MRI which are performed on the 0.2% of MRI systems that operate at 7T or greater field strengths.

**Author Contributions:** Conceptualization, P.C., D.M.C., M.S., T.R.M., M.B., E.E.K., J.P.M., G.G.H., M.T. and G.M.B.; methodology, P.C. and G.M.B.; software, D.M.C. and M.B.; validation, P.C., M.B. and G.M.B.; formal analysis, P.C.; investigation, P.C., D.M.C. and T.E; resources, G.N. and S.J.H.; data curation, P.C., G.N., T.E. and M.T.; writing—original draft preparation, P.C. and G.M.B.; writing—review and editing, all; visualization, P.C., D.M.C., J.P.M., G.G.H., M.T. and G.M.B.; supervision, S.J.H. and E.E.K.; project administration, S.J.H. and E.E.K.; funding acquisition, G.N. and S.J.H. All authors have read and agreed to the published version of the manuscript.

**Funding**: We would like to acknowledge support from the Fred Lewis Enterprise Foundation, the Hugh Green Foundation, Manaakia Moves Trust, the JN & HB Williams Foundation, Mangatawa Beale Williams Memorial Trust, and Kānoa New Zealand.

**Acknowledgments:** We would like to acknowledge support from the Fred Lewis Enterprise Foundation, the Hugh Green Foundation, Manaaki Moves, NZ P Pull, the JN & HB Williams Foundation, the Mangatawa Beale Williams Memorial Trust, Anonymous Donor, Neurological Foundation, Trust Tairāwhiti, Friends of Mātai Blue Sky Fund, GE Healthcare, Mātai Ngā Māngai Māori, and Kānoa—Regional Economic Development & Investment Unit of New Zealand.

**Conflicts of Interest:** All authors have completed the ICMJE uniform disclosure form (available at http://dx.doi.org/10.21037/qims.2020.04.07 (accessed on 1 May 2024). The authors have no conflicts of interest to declare.

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
