# Peer review of "Ultra-High Contrast MRI: The Whiteout Sign Shown with Divided Subtracted Inversion Recovery (dSIR) Sequences in Post-Insult Leukoencephalopathy Syndromes (PILS)"

_tomography, doi:10.3390/tomography10070074_

Round 1

Reviewer 1 Report

Comments and Suggestions for Authors

In this study the authors described the concept of ultra-high contrast MRI, the whiteout sign, the theory underlying the use of dSIR sequences and post insult leukoencephalopathy syndromes. Some concerns are listed as below:

Some traditional neuroimage scans can be removed from the manuscript (too many figures in this manuscript).

The structure of this study should be revised (not easy to follow).

Statistical analysis data are lacking in this study.

Comments on the Quality of English Language

fine

Author Response

Thank you for your succinct analysis.

(i)  We apologize for the traditional neuroimages but this is an educational review and potential readers may include physicians and surgeons who have not had a formal radiological training. They may not be familiar with basic concepts such as contrast or the history of neuroradiology. They may also be unaware of the exceptional efforts that were required to create useful contrast in neuroradiology before the advent of CT and the fact that the brain was only indirectly shown previously.

The layout of the paper appears poor to us and we have asked that the historical images (Figs. 1-12) be much reduced in size rather than be eliminated, because it may be much easier for non-neuroradiologists to understand the history of brain imaging and the need for contrast pictorially, rather than through textural descriptions.

(ii)  Yes, we have revised the structure of the paper in accordance with your views to reduce fragmentation, decrease indexing and utilize bullet points.

(iii)  With respect to statistics, there are only two cases included in the paper and these are primarily to illustrate technical points so we have not used clinical statistics.

Reviewer 2 Report

Comments and Suggestions for Authors

Reviewer suggestions

The article discusses Ultra-high contrast (UHC) MRI refers to types of MRI where, using UHC techniques, very high contrast is observed, but little to no contrast is detected on traditional MRI pictures. Using the split subtracted inversion recovery (dSIR) sequence, one of these methods can yield ten times the contrast of the basic inversion recovery (IR) sequences in modeling experiments.

The article has further discussed the concept of UHC-MRI, the whiteout sign, the theory underlying the use of dSIR sequences, and post-insult leukoencephalopathy syndromes

Recommendation

1.      The review work is original and has scientific potential.

2.      The paper is written well and discusses each parameter well.

3.      All the figures are discussed well in the main text of MS.

4.      References are specific and sufficient.

Minor comments

1.      If figures are adopted from the published data or website, check and cite them properly in legend.

2.      The subsections 5.2.1, 5.2.2, and so on, do not have any sub-heading. It's better to have a bullet than a number.

3.      Cite the specific and suitable references in the discussion section, especially where you have discussed figures.

4.      Your group's recently published paper ‘Ultra-High Contrast MRI: Using Divided Subtracted Inversion Recovery (dSIR) ………Study the Brain and Musculoskeletal System’ is relevant to this review, you can cite this article in revised MS.

Author Response

(i)  Thank you for your recommendations including your opinion that the paper is well written and discusses each parameter well. We also appreciate your comments that the figures are discussed well and that the references are specific and sufficient.

(ii)  Many of the first 12 figures are well known and are in the public domain. The paper is an educational review and they are included for the benefit of those who have not had a formal radiological training and may be unfamiliar with the concept of contrast or the means that were necessary to create it in the past in order to demonstrate disease of the brain. The historical images (Figs. 1-12) appear far too large in the text and we have asked that they be much reduced in size. We have included references in the Legends where appropriate.

(iii)  Yes, we eliminated the excessive indexing and used bullet points as suggested by Reviewer 1.

(iv)  Yes, we have included more referencing in the Discussion.

(v)  Thank you. We have included a reference to the recent paper of ours which you cited (ref 20).

Reviewer 3 Report

Comments and Suggestions for Authors

Ultra-High Contrast MRI: The Whiteout Sign Shown with Divided Subtracted Inversion Recovery (dSIR) Sequences in Post Insult Leukoencephalopathy Syndromes (PILS)

The authors can find my suggestions, concerns, and recommendations, section-by-section, as follows:

In the first two sections, I did not note anything incorrect and frankly speaking, I have appreciated both the historical excursus and description of the theory underlying dSIR. According to me, the figures and associated captions are explicative and well written. Similarly, the English language is excellent and I have nothing to write about.

I advise removing the brief sentence describing the content of each section at the beginning.

In section 2.1, I recommend avoiding the constant recall of the figures. This makes difficult the reading.

I found the Case 1 very interesting. Did the authors perform a neuropsychological assessment? Did the patient show specific psychiatric symptoms?

Case 2: The authors did not report specific GCS assessments. Could you add more information about?

Discussion:  The authors stated : “The initial part of the discussion is organized around (i) ultra-high contrast MRI; (ii) the 406 whiteout sign; (iii) dSIR sequences and (iv) post insult leukoencephalopathy syndromes 407 (PILS). Following this, other issues are reviewed.” I advise removing these sentences, and I suggest to add a brief paragraph about the case-series results and then an introduction to iii and iv. Please, correct and rewrite.

Section 5.1 is interesting and well-written. However, the sections 5.2 and 5.3 need to be rewritten. It is quite hard to read and integrate the info in the micro-subparagraphs. Moreover, I recommend adding specific references in the text. The same for 5.4, 5.7  and 5.9.

Please, rewrite the fragmented discussion 

Author Response

(i)  Thank you for your appreciation of the historical excursions and description of the theory underlying dSIR in our paper. Also we appreciate your comments about the figures and associated captions, as well as the quality of the English language.

(ii)  Thank you also for your comment about the brief sentences at the beginning of each section. The paper is an educational review and we have been asked in previous submissions of papers of this type to include introductory comments at the beginning of sections to guide readers who may be unfamiliar with the subject. We have modified the introduction to the Discussion and removed our introductory comments.

(iii)  Yes, we agree the recall of figures may be tedious but we wished to be quite specific about the features of the figures that we drew attention to.

(iv)  In the interests of brevity we did not include the full history of Case 1 who was treated by a neuropsychiatrist (Gil Newburn, co-author on the paper). Detailed neuropsychological assessments were performed. We have added an overview of these to the text.

(v)  Case 2 did not regard his symptoms post injury as sufficient to require medical attendance at a hospital/clinic and he was only studied 21 hrs post injury because he was enrolled in a separate research study. He had no GCS assessment at the time of his first MRI examination (21 hrs post injury) and was asymptomatic at the time of his second examination (64 hrs post injury).

(vi)  As above, we have changed the introduction to the Discussion

(vii)  Yes, the Discussion was fragmented. We have reorganized it, eliminated much of the indexing and many of the subheadings, and used bullet points as recommended by Reviewer 1.

Round 2

Reviewer 1 Report

Comments and Suggestions for Authors

The authors have addressed my previous concerns.

Comments on the Quality of English Language

fine

Reviewer 3 Report

Comments and Suggestions for Authors

The Authors addressed my concerns. I agree with the authors that this manuscript was thought to be a useful guide for students and MR technicians.